# High-resolution single-cell 3D-models of chromatin ensembles during Drosophila embryogenesis

Qiu Sun [1,4], Alan Perez-Rathke [2,4], Daniel M. Czajkowsky [3], Zhifeng Shao [3✉] & Jie Liang [2✉]

Single-cell chromatin studies provide insights into how chromatin structure relates to functions of individual cells. However, balancing high-resolution and genome wide-coverage remains challenging. We describe a computational method for the reconstruction of large 3D-ensembles of single-cell (sc) chromatin conformations from population Hi-C that we apply to study embryogenesis in *Drosophila*. With minimal assumptions of physical properties and without adjustable parameters, our method generates large ensembles of chromatin conformations via deep-sampling. Our method identifies specific interactions, which constitute 5–6% of Hi-C frequencies, but surprisingly are sufficient to drive chromatin folding, giving rise to the observed Hi-C patterns. Modeled sc-chromatins quantify chromatin heterogeneity, revealing significant changes during embryogenesis. Furthermore, >50% of modeled sc-chromatin maintain topologically associating domains (TADs) in early embryos, when no population TADs are perceptible. Domain boundaries become fixated during development, with strong preference at binding-sites of insulator-complexes upon the midblastula transition. Overall, high-resolution 3D-ensembles of sc-chromatin conformations enable further in-depth interpretation of population Hi-C, improving understanding of the structure-function relationship of genome organization.

[1] Shanghai Center for System Biomedicine, Shanghai Jiao Tong University, Shanghai 200240, China. [2] Department of Bioengineering, University of Illinois at Chicago, SEO, MC-063, Chicago, IL 60607-7052, USA. [3] School of Biomedical Engineering, Shanghai Jiao Tong University, Shanghai 200240, China. [4] These authors contributed equally: Qiu Sun and Alan Perez-Rathke. ✉email: zfshao@sjtu.edu.cn; jliang@uic.edu

Understanding the principles of genome organization is essential for gaining insight into fundamental biological processes such as gene expression and DNA replication[1–3]. Studies based on chromosome conformation capture (3C) and related techniques (4C, 5C and Hi-C) have uncovered important structural features of genome folding[4–7]. Among these, topologically associating domains (TADs) are structural units within which chromatin exhibits more frequent interactions[8–10]. They are prominently found in Hi-C studies of many cell types[9] and across different species[10]. In addition, insulator complexes are found to be preferentially located at boundaries of TADs in multiple species[10–12]. Despite abundant high-resolution information provided by Hi-C studies, direct knowledge on how the physical structure of the chromatin determines the relevant cellular functions in individual cells could not be simply inferred due to the intrinsic population-averaged nature of these studies. Important questions such as how chromatin conformations of TADs appear at the single-cell level also cannot be resolved based on such data alone.

Recent single-cell Hi-C studies showed that many structural features defined by population Hi-C such as TADs and compartments vary among individual cells[13–16]. 3D fluorescent in situ hybridization (FISH) and super-resolution imaging technologies further revealed various details of TAD-like structures in individual nuclei[17–20]. A recent study with S2R+ cells, which are derived from late embryos of *Drosophila*, revealed the well-organized nature of repressed TADs in single cells[19]. At the nanometer-scale, single-cell TAD-like structures with sharp boundaries are found to be widely distributed along the human genome[18]. A common finding from these single-cell studies is the large variability of chromatin structures among individual cells.

Despite rapid progress, there are limitations to present single-cell approaches. Although single-cell Hi-C captures chromatin contacts and can characterize cell-to-cell variability, it is difficult to obtain detailed chromatin structures at high resolution owing to the highly sparse nature of single-cell Hi-C data[21]. In contrast, super-resolution imaging studies can indeed provide fine-resolution information on single-cell genome folding, but are restricted to limited coverage and a moderate number of cells.

Due to these limitations, our understanding of important aspects of chromatin organization remains incomplete. While population Hi-C studies can identify a large amount of chromatin contacts, it is unclear to what degree they are physically required for the formation of structures such as TADs[22,23]. It is also unclear, among spatial neighboring relationships identified in single-cell studies, which relationships reflect functional associations and which are due to random collision owing to volume confinement and other factors[24–26]. Furthermore, while 2D frequency maps from population Hi-C exhibit highly detailed patterns, the heterogeneity of the chromatin 3D structures in the underlying cell population could not be quantified. In addition, whether there are a small number of driver interactions of biological importance that are determinants of chromatin folding remains unanswered[27–29].

In this study, we describe a computational approach providing a unified and consistent model that simultaneously: (1) uncovers a small set of important specific interactions from Hi-C measurements that are not due to random collision, exhibit novel biological patterns, and can drive chromatin folding; (2) provides large ensembles of 3D chromatin conformations generated from the specific interactions that largely reproduces population Hi-C measurements; (3) quantifies the heterogeneity of 3D chromatin structures in the cell population; and (4) provides detailed information on chromatin architectures in models of single-cell chromatin conformations.

We apply this method to study the 3D chromatin structures in *Drosophila* cells at different stages of embryogenesis. Examination of modeled single-cell chromatin conformations reveals a number of novel insights. We find that in a representative region of 1 Mb, TAD-like structures exist in >50% of pre-MBT (midblastula transition) cells with boundaries at varying locations, even though the corresponding population Hi-C maps have no TAD structures that can be detected with confidence and are essentially featureless. The boundaries become more fixated at later developmental stages, with strong preference for binding sites of insulator complexes. In addition, the overall heterogeneity of chromatin conformations is significantly reduced at later developmental stages. This is accompanied by dramatic changes in 3D measurements of chromatin compactness. Furthermore, a functional unit of three-body interactions is found to exhibit stage-dependent structural changes.

Overall, our method can transform population Hi-C into 3D models of single-cell chromatin conformations at high resolution. It quantitatively connects statistical patterns in Hi-C maps to physical 3D chromatin structures. Our method complements current single-cell techniques, as it can be used to re-interpret more abundantly available population Hi-C data, while not being restricted in either genome coverage or the number of single cells. Novel biological findings can be gained quantitatively through analysis using our method, which can help to answer important questions such as the relationship between genome structure and genome function.

## Results

**Overview of our approach to 3D chromatin modeling.** Our method relies on the recently developed capability of deep sampling to generate 3D chromatin ensembles[25,28,30] (Fig. 1). We first identify a set of *specific chromatin interactions* from deep-sequenced population Hi-C data. They are unlikely due to ligation of randomly collided loci and are identified by comparing measured Hi-C frequencies to those simulated from an ensemble of randomly folded 3D chromatin configurations. The specific interactions identified are then used as restraints to generate an ensemble of modeled single-cell chromatin configurations for either a genomic region (200 kb to 4 Mb at 2 kb resolution) or a whole chromosome (22 Mb at 5 kb resolution). Structural analysis, including Euclidean distance measurement and spatial

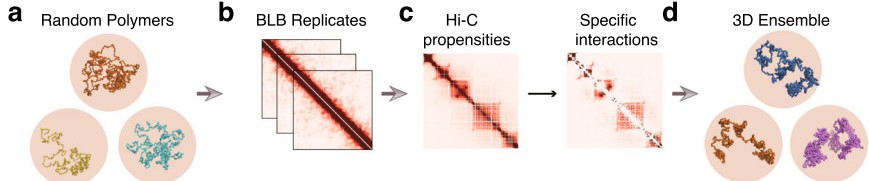

**Fig. 1 Overview of our chromatin modeling strategy. a** We first construct an ensemble of random chromatin polymers as our physical null model. We then **b** bootstrap the polymer ensemble to compute statistical significance of each Hi-C interaction. **c** After removal of non-specific interactions, **d** we construct an ensemble of 3D chromatin structures constrained by specific interactions we identified under a sequential Bayesian inference framework.

clustering, is then carried out on the ensemble of modeled single-cell chromatin configurations. This provides an overall quantitative assessment of population 3D structural properties, such as compactness, radius of gyration, domain boundaries, as well as chromatin structural heterogeneity. Results from structural analysis are then integrated with other information such as enhancer/promoter information and epigenetic modifications for additional biological insights.

**Modeling uncovers a small set of specific interactions.** It has been well known that Hi-C measurements may contain many spurious contacts that simply result from random collisions of chromatin fibers confined inside the nuclear volume[24,28]. Yet these data also contain specific contacts that play important roles in generating the 3D chromatin structures. To distinguish between these two, we sought a means to identify the random contacts by using a null model based on biophysical simulation of random chains to remove the random contacts. Specifically, our null model, which has no dependence on the Hi-C data for its construction, is that of a large ensemble of random, self-avoiding, 3D polymer chains of chromatin fiber in nuclear confinement. Extending the technique of C-SAC[25], we use the fractal Monte Carlo method[30] (see Supplementary Methods) to construct our null models. In this work, we focus on Hi-C data from three *Drosophila* cell types at different developmental stages, namely, embryos pre Mid-Blastula Transition[31] (pre-MBT, cycles 9–13, or stage 3–4), post-MBT[31] (stages 5–8), and S2R+ derived from late embryos[12], to characterize the changes in specific contacts during embryogenesis.

Briefly, for our null model, we generate multiple chromatin chains through a fractal Monte Carlo method based on chain growth[25,28,30,32,33], in which monomers are added one at a time. To overcome severe difficulties in sampling, this process is optimized through a recursive resampling algorithm applied at predetermined check-point lengths[25,28,30]. The target distribution of the ensemble is over all geometrically realizable self-avoiding chromatin chains within the specified volume confinement. This target distribution is rigorously enforced through proper importance weighting during chain growth (see Supplementary Methods for more details).

For each cell type, the null model is an ensemble of $2.5 \times 10^5$ chromatin chains of 4 Mb length, each consisting of 2000 beads at 2 kb resolution. The constraining volume is proportional to the nuclear volume of each cell type (Fig. 1a, see Methods).

We estimate the random contact probabilities of pairs of loci by counting the frequency of 3D conformations in which the spatial distance of the corresponding pair of beads is within a distance threshold of 80 nm[27,28]. We evaluate the statistical significance of each Hi-C contact by bootstrapping the corresponding random ensemble (Fig. 1b, see Methods). Hi-C interactions with BH-FDR adjusted *p*-values below a threshold of 0.01 are then identified as specific interactions (Fig. 1c and see Supplementary Methods).

Based on this criterion, we are surprised to find that only a very small minority of all Hi-C contacts are specific: there are $2.28 \times 10^6$ out of $42.39 \times 10^6$ counts of contact pairs (5.4%), $2.03 \times 10^6$ out of $40.07 \times 10^6$ (5.1%), and $2.20 \times 10^6$ out of $34.98 \times 10^6$ (6.3%) Hi-C contact pairs that are specific for the embryos at pre-MBT, post-MBT, and S2R+, respectively.

**Specific interactions recapture known long-range interactions.** While few, we find that specific interactions contain highly significant information. This can be clearly seen in the heat maps of a polycomb-repressed region shown in Fig. 2a. Despite the fact that only ~5% of Hi-C interactions are retained, key structural

patterns such as the progressive formation of TADs and their finer structures are all present (Fig. 2a, Supplementary Fig. 3a for other regions).

To further evaluate the specific interactions identified, we examined whether long-range interactions with known biological functions are present in this data. In particular, we examined the gene *Bsg25A*, which is transcribed during the minor wave of zygotic genome activation (ZGA)[34], and forms a long-range interaction with the gene, *slam*, in early embryos[31]. This interaction is indeed identified as a specific interaction in our model as seen in the virtual 4C plot (Fig. 2b, lower half), which appears to have captured all relevant long-range interactions, while containing much less noise (Fig. 2b, upper half). Another known long-range loop interaction in late embryos is between gene *Scyl* and *chrb*[31,35], and this is also identified as a specific interaction (Supplementary Fig. 3b).

**Specific interactions reveal biologically-relevant patterns.** The predicted specific interactions also enable a more reliable identification of the biologically relevant trends in the genomic contacts, which would otherwise be obscured. This is illustrated by an analysis of the genome-wide epigenetic properties of the specific interactions. Genomic regions of *Drosophila* can be classified broadly into four chromatin states[12,36] by clustering signals of 15 histone modifications and other biomarkers (Supplementary Figs. 3c and 5, Supplementary Tables 2 and 3). These states are: Active (A), Inactive (I), Polycomb-repressed (P), and Undetermined (U) (Supplementary Fig. 3c). The global distribution of interaction types among specific interactions exhibits overall an increasing number of Inactive–Inactive (I–I) and Polycomb–Polycomb (P–P) contacts in later embryos of post-MBT (stages of 5–8) and S2R+, while Active–Inactive (A–I) and Active–Polycomb (A–P) interactions are found to have steadily declined (Fig. 2c). In contrast, the global distribution of interaction types of all nonzero Hi-C interactions in the original data exhibit no clear pattern: there are only small and random variations among cells at different stages of embryogenesis, regardless the interaction types (Fig. 2d).

In addition, we also find that the density curves of specific interactions at different genomic distances exhibit increased TAD-level (≤400 kb) contact frequencies in embryos at post-MBT (stages 5–8) and S2R+ (Supplementary Fig. 3d, top). In contrast, no such pattern can be found in density curves of all Hi-C interactions (Supplementary Fig. 3d, bottom). Thus, the set of specific interactions in particular is in close agreement with observations that TADs are established progressively during *Drosophila* embryogenesis[31,37].

**Specific interactions can drive chromatin folding.** To assess the roles of specific interactions, we asked whether they alone can drive chromatin to fold into conformations as measured in Hi-C studies. To ensure that our conclusion is general, we examined 10 genomic regions of varying lengths (200 kb to 2 Mb, Supplementary Table 1).

We constructed 3D ensembles of single-chain chromatin conformations at high resolution of 2 kb using the frequencies of the specific interactions. The conformations are generated through a novel approach of sequential Bayesian inference (Fig. 3a–b, see also Supplementary Methods and Fig. 1).

For each region, we construct a Hi-C concordant ensemble of 50,000 single-chain conformations. To constrain the ensemble according to the Hi-C interactions, we derive contact probabilities from the Hi-C frequencies. To estimate this contact probability, we take the minimalistic assumption that DNA fragments in close proximity are available for Hi-C ligations. The contact

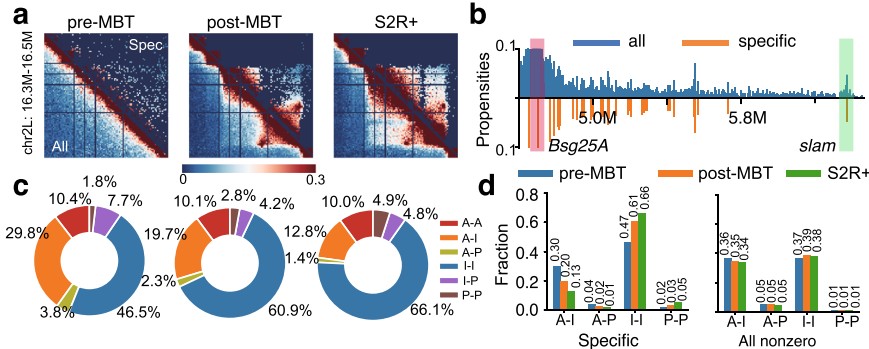

**Fig. 2 Overview of specific Hi-C interactions. a** Heat maps of specific interactions identified in a polycomb-repressed region (chr2L: 16.3–16.5 Mb) of three cell types at different developmental stages. Lower-left triangles represent all Hi-C interactions, upper-right triangles represent the identified specific interactions. Cell types from left to right are embryos at pre-MBT stages (cycles 9–13), embryos at post-MBT (stages 5–8), and S2R+, respectively. **b** A virtual 4C plot of the distribution of specific interactions in a 1.86 Mb region of embryos at pre-MBT stages (cycles 9–13). The red bar represents the anchor which contains the gene *Bsg25A*, and the green bar represents a specific interaction that targets the gene *slam*. **c** Pie charts of percentages of different types of specific interactions in the three cell types. A: active, I: inactive, P: polycomb-repressed. I–I interactions increase from 46.5% to 60.9% and then to 66.1%, P–P increases from 1.8% to 2.8% and then to 4.9%, while A–I decreases from 29.8% to 19.7%, then to 12.8%, and A–P decreases from 3.8% to 2.3%, then to 1.4%. **d** Percentages of four interaction types of specific interactions (left) and all nonzero Hi-C interactions (right). These four types are A–I, A–P, I–I and P–P, respectively. Source data are provided as a Source Data file.

probabilities are then taken as the proportions of 3D conformations in which the spatial distances of the specific pairs of loci of interest are within the distance threshold of 80 nm. Thus, the target distribution of the chromatin ensemble is that of self-avoiding chromatin chains that satisfy the Hi-C derived contact probabilities at each chain growing step, where beads are placed sequentially for each chain. Our sequential Bayesian inference framework ensures that the chromatin chains are consistent with the observed Hi-C, and ensemble properties can be accurately estimated (Supplementary Fig. 7). We then aggregate single-chain conformations to obtain the simulated Hi-C contact maps (Fig. 3c, Supplementary Fig. 8a).

As shown in Fig. 3d, simulated Hi-C contact maps using only specific interactions consistently exhibit strong similarities to measured Hi-C contact maps across the 10 regions (Supplementary Fig. 8, Pearson correlation coefficients $r = 0.91$–$0.98$, distance-adjusted correlation coefficients $r' = 0.56$–$0.81$[38]). Moreover, the log-log scaling curve of simulated contact probabilities at different genomic distances and the corresponding curve from Hi-C largely match each other (Fig. 3e).

To validate such results over larger genomic regions, we then constructed an ensemble of 3D chromatin conformations for the whole X chromosome of S2R+ cells at 5 kb resolution, using 4,485 beads and only 6.1% of the Hi-C contact pairs that are considered to be specific (Fig. 3h). Figure 3i shows two examples of the conformations of the X chromosome. Again, the simulated and measured Hi-C maps are highly concordant ($r = 0.94$ and $r' = 0.64$).

Overall, these results demonstrate that using only 5–6% of measured Hi-C contacts that are predicted as specific interactions, we can consistently reproduce experimental Hi-C contact maps across different regions in a chromosome with high accuracy at high resolution. This strong agreement is maintained at the whole chromosome level. Our results thus demonstrate that predicted specific interactions are sufficient to drive chromatin folding in *Drosophila*.

**Specific interactions recover 3D loops with improved clarity.** To further ascertain the roles of specific interactions, we compare simulated ensembles of chromatin chains for the 10 different 4-Mb regions using only specific interactions, or all Hi-C interactions, or only non-specific interactions of the same number of

contact pairs as the specific interactions (Fig. 3c). Simulated heat maps of contact probability using all or specific interactions exhibit strong similarities to the corresponding heat maps of the measured Hi-C ($r = 0.92$–$0.98$, $r' = 0.56$–$0.81$ and $r = 0.91$–$0.98$, $r' = 0.58$–$0.81$ respectively, and see Supplementary Table 1). In contrast, simulated Hi-C heat maps using non-specific interactions fail to capture much of the structural features observed in the Hi-C maps ($r = 0.48$–$0.58$, $r' = -0.02$–$0.27$, Fig. 3c and d).

We note that although ensembles generated using all Hi-C contact frequencies and using only specific interactions have similar correlations with the measured Hi-C, the latter can recover structural features such as loops with better clarity (Fig. 3c). This is illustrated by the detailed height maps of contact probability of the simulated 3D chromatin chains in a ~40 kb × 40 kb region, where a loop interaction site is located (Fig. 3g). The height map of contact probability calculated from the ensemble by specific interactions has a much stronger resemblance to the original Hi-C frequency map (left) than that from the ensemble by all Hi-C contacts. The control ensemble from non-specific interactions fails to capture this loop structure.

As an additional measure of the effectiveness with which loops are identified, we examined the spatial distances between the two anchors of this loop interaction (Fig. 3f). From the ensemble generated by specific interactions, 41.5% of conformations from the ensemble by specific interactions have a spatial distance less than the ligation threshold of 80 nm. In contrast, the percentage is only 20.0% and 1.7% for ensembles generated using all Hi-C and non-specific interactions, respectively. For a pair of loci in a nearby region and of the same genomic distance but without looping interaction, the fractions of conformations with the loci within the ligation threshold are indistinguishable between ensembles from specific and from all interactions (11.1% and 12.4%, Supplementary Fig. 8c).

These results show that chromatin ensembles reconstructed by specific interactions have better structural clarity in defining loop interactions without compromising detection specificity.

**Single-cell conformations quantify chromatin heterogeneity.** Previous studies showed that chromatin organization is established progressively during *Drosophila* embryogenesis[31,37]. A number of long-range interactions are found to exist during the

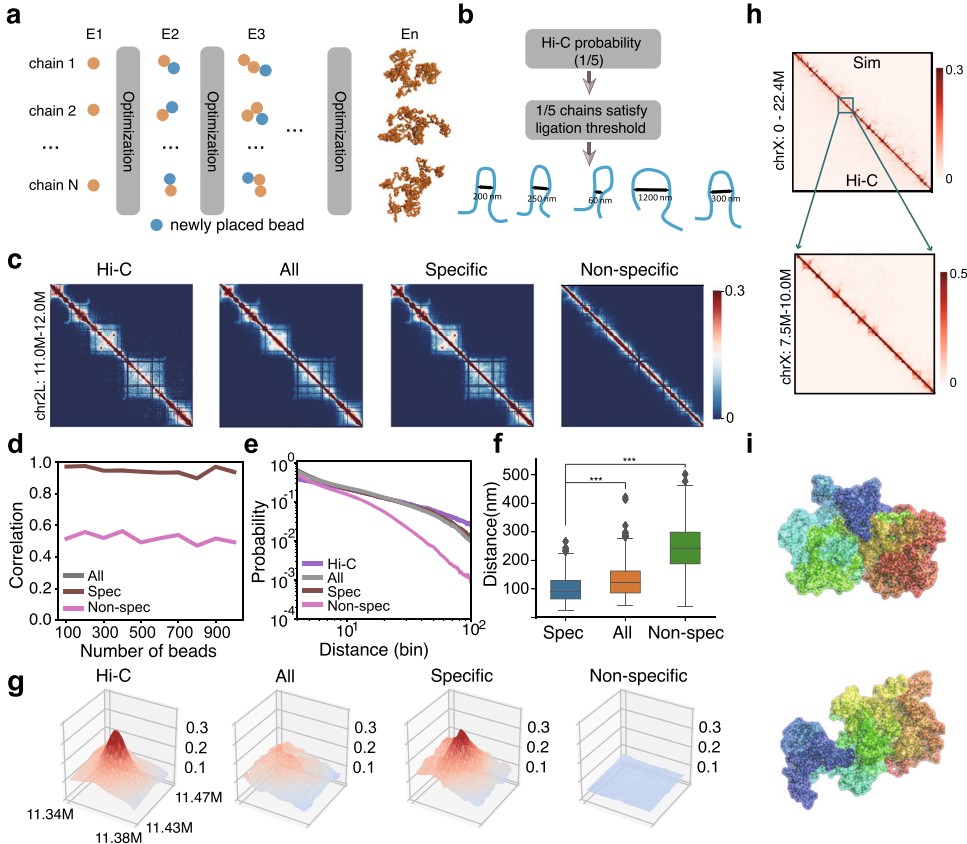

**Fig. 3 Specific interactions are sufficient to drive chromatin folding in *Drosophila*. a** The chromatin polymer ensemble is constructed sequentially, with the addition of one bead at a time. We make use of an optimized sampling distribution at each step to improve sampling efficiency, which is dynamically adjusted after adding each bead. **b** In our model, Hi-C contact probability corresponds to the proportion of polymer chains that satisfy the ligation threshold. **c** Illustration of simulation results for a 1 Mb region (chr2L: 11.0–12.0 Mb) of S2R+ cells at 2 kb resolution. Heat maps from left to right represent Hi-C propensities, simulated contact probabilities using all, specific, and non-specific interactions, respectively. **d** Pearson correlation coefficients of the simulated contact probabilities and Hi-C propensities in 10 regions of different genomic lengths. The number of beads ranges from 100 to 1000. **e** log-log scaling curves of contact probabilities with genomic distances (bin) derived from the original Hi-C data, simulated ensembles using specific, all, and non-specific interactions. **f** Distance distributions of the two anchors of the loop shown in **c**. Loop anchors correspond to bead No. 170 and bead No. 190. *** represents two-sided Wilcoxon rank sum test *p*-value ≤ 0.001. **g** Height maps of contact probability of the loop interaction inside the polycomb domain shown in **c**. **h** Constructed 3D conformations of chromosome X using specific interactions at 5 kb resolution. (Top) Simulated Hi-C heat map, with measured Hi-C propensities at the bottom and simulated contact probabilities at the top. (Bottom) Zoomed-in heat map of a 2.5 Mb region. Pearson correlation coefficient *r* is 0.94, distance-adjusted correlation *r'* is 0.64. **i** Visualization of two examples of 3D conformations of chromosome X using PyMOL. Source data are provided as a Source Data file.

minor wave of zygotic transcription at pre-MBT cycles 9–13[34]. This is followed by the establishment of TADs and compartments during the midblastula transition (MBT)[39]. These TADs are clearly discernible in Hi-C profiles of embryos at stages 5–8 and S2R+ during and after MBT (Fig. 2a). However, these findings are all based on two-dimensional analysis of population Hi-C, and their implications in terms of the actual 3D folding of the chromatin is not clear.

Therefore, we set out to examine the 3D structural changes of chromatin during *Drosophila* embryogenesis. Specifically, we examined a polycomb-repressed region and an active region of the same length in cells at the different developmental stages, namely, embryos at pre-MBT (cycles 9–13), post-MBT (stages 5–8), and S2R+. Overall, simulated heat maps of both regions show strong similarities to the corresponding heat maps of measured Hi-C (Supplementary Figure 9a for the repressed region and S9b for the active region, $r = 0.95$–$0.98$, $r' = 0.71$–$0.76$).

As our simulated ensembles contain $5.0 \times 10^4$ properly-sampled single-cell chromatin conformations (Fig. 3i), we are able to quantify the heterogeneity of the modeled cell population, which would not be possible for chromatin models based on a single consensus structure such as those in[40–42]. Using hierarchical clustering[43], we grouped the modeled single-cell chromatin conformations of all three stages of cells into 5 clusters (Fig. 4a and b, see Supplementary Methods). We observe significant differences among the different clusters of the 3D single-cell conformations of the polycomb-repressed region in early embryos of pre-MBT, while the heterogeneity reduced somewhat in later embryos. In S2R+, two clusters dominate, accounting for 41.1% and 55.7% of the conformations, respectively. Interestingly, the largest cluster (C5) is the same for pre-MBT (31.7%) and S2R+ (55.7%), but a different cluster (C4) transiently become the largest cluster (50.1%) at post-MBT (Fig. 4c). The active region also exhibits significant heterogeneity among single-chain chromatin conformations (Supplementary Fig. 9e and 9f). Thus, there are similarities as well as notable differences in the modeled conformations of subsets of cells at different stages of embryogenesis, which are not apparent from the population-level Hi-C maps.

**Increasing compactness of polycomb regions in late embryos.** Consistent with previous qualitative observations of significant

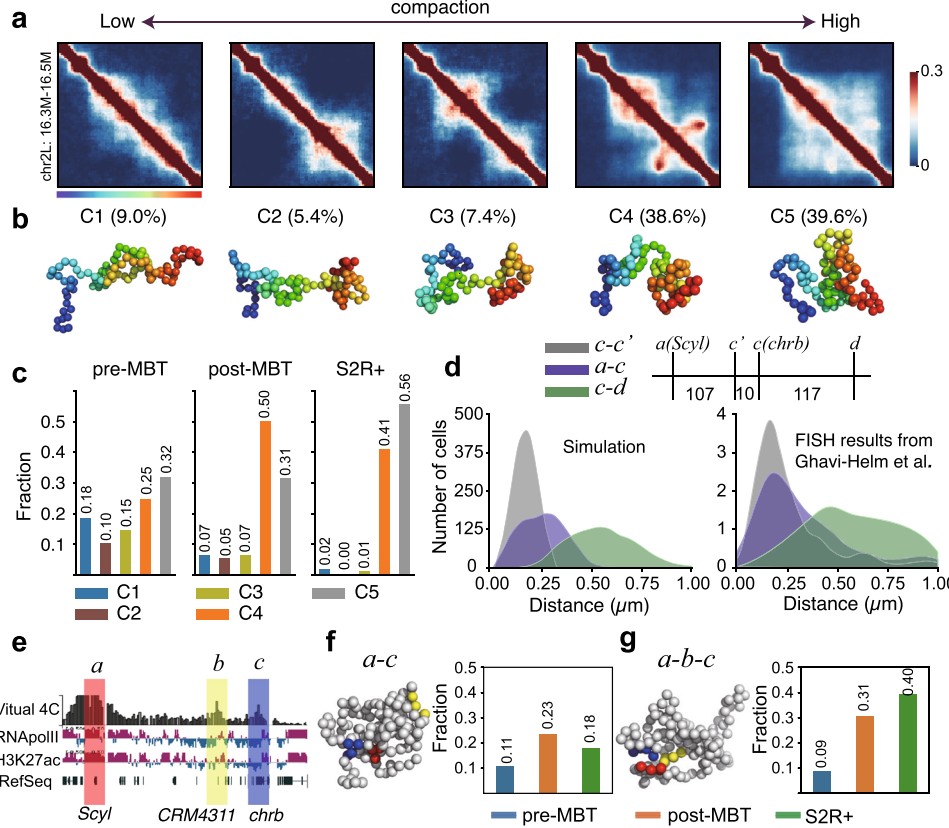

**Fig. 4 Chromatin heterogeneity, compaction, and dynamic changes of a 3-body interaction unit. a** Hierarchical clustering of 3D chromatin configurations of all three stages of cells in the polycomb-repressed region. Aggregated contact heat maps of the 5 clusters are ordered based on their averaged radius of gyration ($R_g$). **b** Representative conformations of the 5 clusters shown in **a**. Proportions are labeled on top of the conformations. **c** Proportions of the 5 clusters in each cell type. pre-MBT: cycles 9–13; post-MBT: stages 5–8. **d** The left figure shows the simulated distance distributions of the a–c (*Scyl–chrb*) interaction and another two control interactions (c–d and c–c'). c–d has the same genomic distance as a–c, while c–c' is a near-range interaction. The right figure shows the distance distributions derived from the DNA FISH measurements[45]. The three interactions are illustrated on the top. **e** A virtual 4C plot of the region that contains a three-body interaction among two promoters and a putative enhancer, with the anchor shown in red bar. The gene *Scyl* is located within this anchor region. Two target regions are shown in the blue bar and yellow bar. The gene *chrb* is localized in the blue region, and a putative enhancer overlapped with a ChIP-defined cis-regulatory module (CRM) *CRM4311* is in the yellow region. Tracks of RNA polymerase II, H3K27ac, and RefSeq genes are shown at the bottom. **f** The left figure shows an example of a two-body interaction between the promoters of genes *Scyl* and *chrb* which are labeled in **e**. The right barplot shows the fractions of this two-body interaction in three different cell types. **g** The left figure shows an example of a three-body interaction among the two promoters and the enhancer labeled in **e**. The right barplot shows the fractions of this three-body interaction among all *Scyl–chrb* interactions in the three cell types. Source data are provided as a Source Data file.

changes in compactness during embryogenesis[31,37], we observed that later embryos have higher proportions of compact clusters (C4 and C5) in the polycomb-repressed region (Fig. 4c). Yet our models further enable a quantification of this difference in compaction: there is an 8% reduction in the radius of gyration ($R_g$)[44] in this region from the pre- to post-MBT but a more significant 18% reduction in the S2R+ cells. Similarly, the end-to-end distances change by only 6% from pre- to post-MBT whereas there is an 18% change in the S2R+ cells (Supplementary Fig. 9c). Similar changes in these measurements are also observed in the active region (Supplementary Fig. 9d and 9i). Such quantitative measures of change in compactness can only be derived from Hi-C and 3C-derived data via physical modeling of 3D chromatin conformations.

**Dynamic changes of functional three-body interactions**. We expected that these changes in the overall compaction at the domain-level might also translate into functionally relevant changes in specific three-body interactions. Genes *Scyl* and *chrb* are involved in head involution and many other biological functions[35] (Supplementary Fig. 6). Previous studies suggested

that the promoters of these genes interact physically with a putative enhancer and this interaction changes during embryogenesis (Fig. 4e)[31,35,37].

To test this idea, we constructed 3D ensembles of single-chain conformations of this region for cell at the three stages. As shown in Fig. 4g, we found there exists a significant three-way interaction between these regions in our models. We note that the distribution of spatial distances between *chrb* and *Scyl* and two other control regions derived from our model is highly consistent with DNA FISH measurements (Fig. 4d)[45]. Our simulations reveal that post-MBT embryos (stages 5–8) have higher *Scyl–chrb* (Fig. 4f) and *Scyl*–enhancer (Supplementary Fig. 9g) contact frequencies when compared to modeled conformations at the other two stages, while *chrb* and the enhancer contact more frequently in S2R+ (Supplementary Fig. 9h). Despite a lower proportion of total *Scyl–chrb* interactions, we find an increased propensity of three-body contact with the putative enhancer among these *Scyl–chrb* interactions in S2R+ (Fig. 4g). Thus, our results implicate this putative enhancer[45] in forming a spatial unit of three-body interactions with the promoters of *Scyl* and *chrb*.

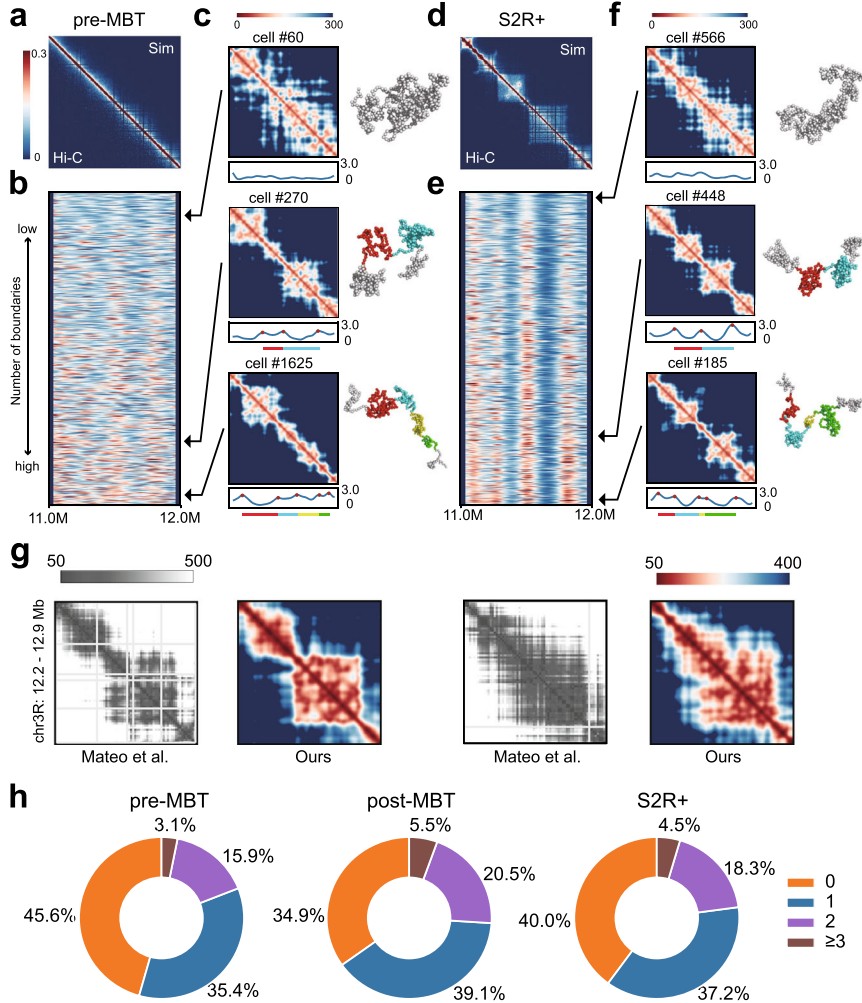

**Fig. 5 TAD-like structures in modeled single-cell conformations during *Drosophila* embryogenesis.** Simulated heat maps of the region (chr2L: 11.0–12.0 Mb) in embryos at pre-MBT cycles 9–13 (**a**) and S2R+ (**d**) are shown. The lower left triangles represent the experimental Hi-C propensities, and the upper right triangles represent the simulated contact probabilities. Resolution is 2 kb. Boundary strength profiles of 5,000 conformations in embryos at cycles 9–13 (**b**) and S2R+ (**e**) are shown below the combo heat maps. They are ordered by the number of domain boundaries. Three representatives of single-cell spatial-distance heat maps (on the left) and corresponding visualizations of conformations (on the right) in embryos at pre-MBT cycles 9–13 (**c**) and S2R+ (**f**) are shown. These single-cell conformations have different numbers of domain boundaries. The number of boundaries from top to bottom is 0, 3, and 5, respectively. Arrows indicate their positions in the boundary strength profiles. Boundary strength curves are drawn under the spatial-distance maps, with red dots representing the local maxima identified as domain boundaries. Bars in different colors under the boundary strength curves represent different domains identified in that conformation, which are also labeled in 3D visualizations. **g** Correspondence between distance maps of two modeled single-cell conformations (chr3R:12.20–12.90 Mb of *Drosophila* S2R+ at 2 kb resolution) with two conformations constructed from imaging studies in Mateo et al. (*r* = 0.75 and 0.79, Fig. 2d of[20] reprinted with permission). Proportions of single-cell conformations with different numbers of TADs are shown in **h** for each cell type. Source data are provided as a Source Data file.

**TAD-like structures in many single-cells of early embryos.** TADs are one of the most prevalent structural units of genome organization that have emerged from Hi-C studies[8–10]. Previous Hi-C studies showed that, during Drosophila embryogenesis, TAD boundaries are only established during the MBT but are otherwise not visible on the heat map at early stages[31,37,39]. Indeed, as shown in Fig. 2a, TADs can be clearly seen in Hi-C heat maps of post-MBT embryos (stages 5–8) and S2R+ cells, but not in early embryos. However, Hi-C heat maps are a reflection of ensemble-averaged properties. This naturally leads to the question of whether TAD-like structures could actually exist in individual cells[21]. Specifically, we ask if there are TAD-like structures in early embryonic cells, and whether single-cell TAD structures are different among cells at different developmental stages.

We examined a 1 Mb region on chromosome 2L, from which several TADs can be found in Hi-C maps of late embryos

(Fig. 3c). The simulated contact maps from the aggregation of $5.0 \times 10^4$ modeled single-cell chromatin conformations are highly similar to the corresponding measured Hi-C maps (Fig. 5a and d, Supplementary Fig. 10a, *r* = 0.97, 0.95, and 0.95 for embryos at pre-MBT cycles 9–13, post-MBT stages 5–8, and S2R+, respectively).

To characterize the chromatin structures of the modeled single-cell conformations, we calculated the spatial distance between each pair of beads and generated a spatial-distance map for each modeled single-cell chromatin conformation, following reference[18] (Fig. 5c and f, Supplementary Fig. 10c). There are clear TAD-like structures in the spatial-distance maps of many modeled single-cell conformations of S2R+ (Fig. 5f), consistent with studies of super-resolution imaging[18–20]. Our modeled single-cell conformations also agree well with single-cell chromatin reconstructed from another imaging study[20] (Fig. 5g

and Supplementary Methods). Surprisingly, we also detect a strong presence of TAD-like structures in many modeled single-cell chromatin conformations of early embryos at pre-MBT cycles 9–13 (Fig. 5c), even though no TADs can be seen in the population Hi-C heat maps (Fig. 5a).

We then examined the domain boundaries in the modeled single-cell conformations. We calculated the boundary strength at each genomic position using a spatial-distance ratio (see Methods) and generated boundary-strength curves (see blue curves in Fig. 5b and e, Supplementary Fig. 10b). We selected single-cell domain boundaries from local maxima above a threshold of 2.2 (in red dots, see Supplementary Methods for details). We identified TAD-like structures from consecutive regions with BH-FDR adjusted $p$-values below 0.05 between pairs of adjacent boundaries (see Supplementary Methods). Finally, we randomly selected 5,000 3D conformations from the ensemble of each cell type in order to identify domain boundaries and TAD-like structures.

We find that a large portion of modeled single-cell conformations contain sharp domain boundaries in early embryos. More than half of the modeled single-cell conformations (54.4%) in embryos at pre-MBT (cycles 9–13) possess at least one TAD-like structure. This is certainly beyond what would be expected from random fluctuations[25] (see also Supplementary Fig. 11a), and is only slightly below that of S2R+ where 60.0% modeled conformations contain at least one TAD-like structure (Fig. 5h). These results suggest that TAD-like structures could exist in individual cells during *Drosophila* embryogenesis, even at the early developmental stage, where TAD structures could not be clearly detected with confidence from population-averaged Hi-C data (see Supplementary Fig. 11b row 1, in contrast to rows 2 and 3[31]).

However, there is strong variability in boundary positions (Fig. 5b and e, Supplementary Fig. 10b) as well as domain sizes (Supplementary Fig. 10d) among individual modeled conformations of early embryos at pre-MBT cycles 9–13. Our results thus suggest that these variabilities are the main reason for the overall absence of concordant TAD structures in the population Hi-C heat maps in early embryos.

**Insulator-binding at predicted single-cell domain boundaries.**
A recent single-cell imaging study revealed that domain boundaries can occur at any genomic positions, but preferentially at CTCF/cohesin binding sites in individual mammalian cells[18]. We calculated the boundary probabilities along the linear genomic positions of the region that is the same as that of Fig. 5a for *Drosophila* cells at the three developmental stages. Results show that while there is no clear position preference for boundaries in early embryos at pre-BMT cycles 9–13, strong preference for specific genomic positions appear in embryos during and after the MBT, at stages 5–8 and S2R+ cells (Fig. 6a). Despite the strong preference, there is a non-zero probability for any genomic position to be at a domain boundary in late embryos, which is consistent with previous findings[18].

Previous studies showed that insulator proteins such as BEAF-32/CTCF/Su(Hw)/CP190 are strongly enriched at TAD boundaries of *Drosophila*[46]. We then examined whether the boundary positions appearing in our 3D models of late embryos also correlate with the preferred binding sites of these insulator proteins. Our results show no enrichment of boundary probabilities at the binding peaks of the insulators of CTCF and cohesin (Fig. 6b). This is consistent with earlier Hi-C studies[12,47]. We then examine binding sites of other insulator complexes with extensive presence in *Drosophila*[12,48]. We find that domain boundaries in 3D models of single chromatin chains of later

embryos are highly enriched at the binding peaks of insulator complexes BEAF-32/CP190 and BEAF-32/Chromator (Fig. 6a and c). Single-cell boundary probabilities are also enriched at binding peaks of ZW5 (Supplementary Fig. 11), which may facilitate specific long-range interactions at the single-cell level[49]. Moreover, our results are consistent with a recent study by Hug et al.[37], where reported boundaries in pre-MBT embryos are associated with genes expressed zygotically before MBT. The boundary probabilities calculated from our predicted ensemble of single-cell conformations for the locus chr2L: 4.5–6.4 Mb are in excellent agreement with data reported in[37] (Supplementary Fig. 11d).

These results therefore show that TAD boundaries appearing in our 3D structural models of single chromatin chains, once aggregated, are consistent with findings from population-based Hi-C studies[46]. Furthermore, the heterogeneity in Hi-C data is manifested by the presence of TADs with strongly heterogeneous boundaries at an early stage, which is quantified in our modeled individual chromatin structures.

## Discussion

The computational method described in this study can quantitatively connect statistical patterns in Hi-C maps to physical 3D chromatin structures. While Hi-C measurements of cell populations have provided a wealth of information on chromatin structures at high resolution[10,12], physical structures of chromatin in individual cells do not automatically follow from such population ensembles. Furthermore, our method can bridge the gap between high-resolution population Hi-C studies and fine-detailed single-cell 3D structures of chromatin[13–20], which are often sparse, with limited coverage and resolution, and are restricted in the number of cells.

Our method identifies specific chromatin interactions from Hi-C measurements, from which biological patterns emerge with clarity. As many interactions in population-averaged Hi-C data are due to experimental biases and random collisions of chromatin fibers in the cell nucleus[24,25,28], it is unclear which ones are required for formation of chromatin structures such as TADs. It is also not known whether all contacts identified in single-cell studies are obligated for TAD formation. By constructing a random model of self-avoiding 3D chromatin polymers in the nuclear confinement (Fig. 1), our method can effectively remove background noises due to random collision. The identified specific interactions exhibit clear biological trends such as increasing Inactive-Inactive and Polycomb-Polycomb genomic interactions during *Drosophila* embryogenesis, which are not seen in the original Hi-C data (Fig. 2c and d).

Our method can be used to gain understanding the relationship between genome structure and function. Results on *Drosophila* cells showed that although the identified specific interactions constitute only a small fraction of measured Hi-C interactions (5–6%), they are sufficient to fold chromatin into an ensemble of conformations exhibiting the full patterns of population Hi-C measurements ($r \sim 0.95$, Fig. 3c and d). It is probable that there may exist even a smaller minimum set of interactions, likely to be functionally important, which are sufficient to drive chromosome folding and give rise to much of the topological features observed in Hi-C[30]. That is, features such as TADs may arise naturally from a small set of functionally important chromatin interactions that are sufficient to drive chromatin folding. This is consistent with a recent study probing genome structure-function relationship, where it was found that functional interactions may play important roles in shaping genome structures[50].

Our results on *Drosophila* amply demonstrate that detailed chromatin model conformations in individual cells can be

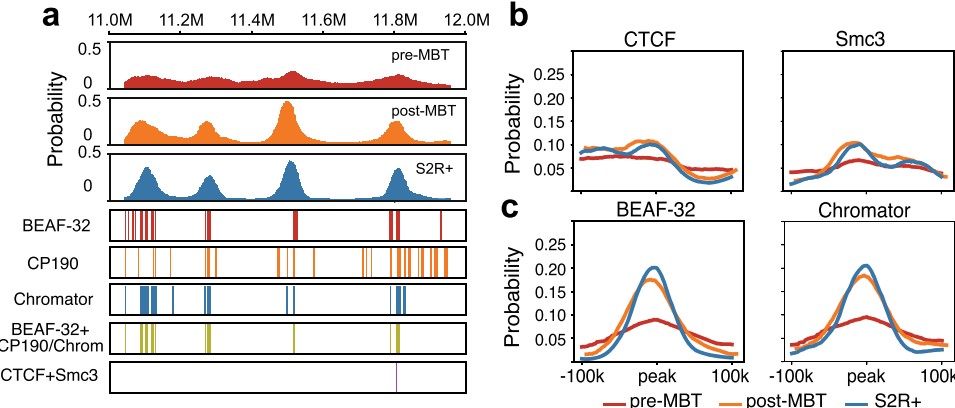

**Fig. 6 Domain boundaries are preferentially localized at insulator binding sites since MBT. a** Distributions of domain boundary probabilities along genomic positions of the same region in Fig. 5A (chr2L: 11.0–12.0 Mb) of three cell types are shown in the top three rows. Tracks in the 4-*th* to the 6-*th* row indicate the binding sites of 3 different insulator proteins of BEAF-32, CP190, and Chromator, respectively. The track in the 7-*th* row indicates the intersection of the binding sites between BEAF-32 and CP190 or between BEAF-32 and Chromator, the last track represents the intersection of the binding sites between CTCF and Smc3 (cohesin subunit). All tracks are from S2-DRSC. **b** Enrichment curves of the averaged domain boundary probabilities at the binding peaks of CTCF and Smc3. **c** Enrichment curves of the averaged domain boundary probabilities at the binding peaks of BEAF-32 and Chromator.

obtained through modeling of population Hi-C data to enable new insight. We found that a large portion (>50%) of modeled single-cell conformations possess TAD-like structures in early pre-MBT embryos (Fig. 5a and c), whereas population Hi-C data are essentially featureless and no clear TAD structures can be seen in previous Hi-C studies[31,37]. Consistent with an earlier finding that domain-like substructures can arise from nuclear volume confinement alone[25], our results show that *Drosophila* chromatin is likely topologically organized to varying extents with TAD-like structures present in single cells throughout different stages of *Drosophila* embryogenesis. Furthermore, we find that the domain boundaries in single cells are preferred at insulator binding sites (Fig. 6a) of the same insulator complexes as found at the TAD borders in population Hi-C analysis[12,48]. As insulator-binding preference is also found in mammalian single cells[18], there may exist a conserved mechanism of genome folding at the single-cell level.

Our method also addresses a long-standing challenge in 3D genome studies, namely, the characterization of chromatin structure heterogeneity. As population-based Hi-C studies offer no directly interpretable information on the heterogeneity of 3D chromatin conformations of the cell population, there is an overall lack of quantitative understanding on the structural heterogeneity of chromatin[51,52]. This prevents us from understanding the actual physical states that the chromatin in each cell must adopt to be functional. With the ability to convert 2D high-resolution population Hi-C heat maps into ~$5.0 \times 10^4$ modeled single-cell chromatin conformations at high resolution, our method can provide quantification of chromatin heterogeneity (Fig. 4c). This allows possible differentiation of chromatin configurations that are functional from those that are not.

Our method is based on an explicitly constructed random 3D polymer ensemble as the null model. This is a uniquely appropriate approach for chromatin studies, as polymer effects of the random collision of chromatin fibers are explicitly modeled. This null model allowed effective discrimination of specific interactions from background noise of random chromatin collision. Several other methods such as Fit-Hi-C[53] and GOTHiC[54] can also be used to identify specific interactions. However, these methods rely on null models constructed from the Hi-C data itself for removal of background noise[53,55,56], which may lead to inherent bias in the resulting null hypothesis test. Furthermore, our physically constructed null model can be used to identify

higher-order specific many-body interactions[30], which is not possible with null models derived solely from 2D Hi-C contact maps. As shown in a separate study describing the CHROMATIX algorithm, this null model can be used for analysis of many-body chromatin interactions, such as those encountered in enhancer-rich regions[30].

We have further compared the specific interactions identified by our method with those by Fit-Hi-C[53] and GOTHiC[54] (Supplementary Fig. 4a and Supplementary Methods). Overall, all three methods identify a small fraction of specific interactions (4.1–7.5% genome-wide for S2R+ cells), with varying degree of overlaps (85.2% of Fit-Hi-C and 36.0% GOTHiC are present in ours). Compared to Fit-Hi-C and GOTHiC, our method identifies more long-range interactions (26.2% vs. 17.3% and 1.35% are of ≥500 kb, Supplementary Fig. 4b–c). The small fractions of interactions identified by all these methods are found to be sufficient to drive chromatin folding for a 1 Mb region tested using our folding algorithm, although ours gives the highest correlation to Hi-C measurements (Supplementary Fig. 4d). We also examined the effectiveness of HiCCUPS[10], which identifies looping interactions. Interestingly, we found that the much smaller fraction (0.1%) of looping interactions identified by HiCCUPS was insufficient to drive chromatin folding (Supplementary Fig. 4).

Our approach for constructing 3D ensembles differs from several existing methods[22,27,57–59] (Supplementary Fig. 2). With only basic physical considerations of fiber density and ligation distance threshold, it is minimalistic and there are no adjustable parameters. No chromatin states are assigned to polymer beads, and there are no a priori assumptions on locations of loop anchors. Furthermore, our method samples well. With deep sampling enabled through Bayesian sequential inference, our method is unique in its ability to generate a large number (e.g., $5 \times 10^4$) of diverse chromatin conformations that are consistent with Hi-C measurements.

Overall, our computational method can quantitatively connect statistical patterns in Hi-C maps to 3D physical chromatin structures. It can quantify chromatin heterogeneity and facilitate discovery of biological patterns at both single-cell and population levels. As demonstrated in the analysis of *Drosophila*, our method can provide new perspectives on genome 3D structural changes during biological processes such as embryogenesis, enabling discoveries that would not be possible with traditional Hi-C analysis. Our method is robust and can be used for arbitrary loci or full

chromosomes. With quality Hi-C data becoming more abundant, applications of our method can aid in overall understanding of the mechanisms of genome folding and can help to decipher the structure-function relationship of genomes.

## Methods

**Hi-C propensities.** Hi-C data are obtained from the GEO database (embryos at pre-MBT cycles 9–13 and post-MBT stages 5–8 from GSE103625, S2R+ from GSE101317) and are mapped to the dm3 reference genome using Bowtie2 (v 2.2.9) following[12]. Hi-C contact matrices generated at 2/5 kb resolution are normalized using ICE from the hiclib[60]. Assuming neighboring regions always form Hi-C ligations[27], Hi-C propensities are calculated as

$$p_{\mathrm{obs}}(i,j) = \frac{C(i,j)}{\mathbb{E}_{diag}(1)} \; , \tag{1}$$

where $C(i,j)$ is the Hi-C contact frequency of loci $i$ and $j$, $\mathbb{E}_{\mathrm{diag}}(1)$ is the averaged contact frequency of Hi-C pairs $(i,j)$ with $|i-j|=1$ bin.

**Model parameters and contact model.** We model random chromatin fibers as self-avoiding polymer chains consisting of beads, each represents a 2 kb or 5 kb genomic region, the same as the resolution of the Hi-C matrices. We assume beads-on-string[61] chromatin fiber has a mass density of 165 bp/11 nm[62], thus the bead diameter is roughly 25 nm. The spherical volume within which the polymer chains are confined are proportional to the nuclear volume of each cell type. We choose 292 μm³ as the volume for S2R+[63], 335 μm³ for embryos at cycles 9–13[64] and 524 μm³ for embryos at stages 5–8[64]. The genome size is approximately 700 Mb for tetraploid S2R+ and 350 Mb for diploid embryos at cycles 9–13 and stages 5–8[65,66].

We assume regions that are in close proximity are available for Hi-C ligation. Contact probabilities are then calculated as the proportions of modeled single-cell conformations satisfying the distance requirement, namely, the distances of the pair of loci of interests are within a threshold, which is the longest distance for ligation.

**Constructing physical null model of chromatin chains.** We generate random chromatin polymer chains using a novel Monte Carlo approach (see Supplementary Methods) and construct an ensemble of $2 \times 10^5$ random polymer chains within a defined space for each type of cell (Fig. 1a). They are used as our null model to estimate contact probabilities $p_{\mathrm{null}}$ of random collisions which lead to non-specific Hi-C interactions. $p_{\mathrm{null}}$ were defined as

$$p_{\mathrm{null}}(i,j) = \frac{\sum_{k=1}^{N}\left[\mathbb{I}^{(k)}(i,j)w^{(k)}\right]}{\sum_{k=1}^{N} w^{(k)}} \; , \tag{2}$$

where $\mathbb{I}^{(k)}(i,j)$ is an indicator function of 1 if the distance between $i$ and $j$ in the $k$-th chain is $<d_c$, with $d_c=80$ nm[27,28], $N$ is the total number of polymer chains, and $w^{(k)}$ is the importance weight of the k-$th$ chain for bias-correction due to deviations of the sampling distribution from the target uniform distribution in our null model.

**Identification of specific interactions.** For each ensemble of random polymer chains of a cell type, we assign a statistical $p$-value to each pair of loci based on the percentage of random contact probabilities in bootstrap replicates that exceed the relative Hi-C propensity. We use *Bag of Little Bootstrap* (BLB)[67] (see Supplementary Methods) to generate a total of 5,000 bootstrap ensembles (Fig. 1b). Although each BLB ensemble contains only a small subset (~1300 polymer chains, or ≤1%) of the original ensemble, the average physical properties of the BLB ensembles reflect that of the original polymer ensemble[67]. For each BLB ensemble, $p_{\mathrm{null}}(i,j)$ is calculated as described above. After quantile normalization of $p_{\mathrm{null}}(i,j)$ and $p_{\mathrm{obs}}(i,j)$, we assign a $p$-value to each pair of loci $(i,j)$ according to the percentage of $p_{\mathrm{null}}(i,j)$ that exceed $p_{\mathrm{obs}}(i,j)$,

$$p_{ij} = \frac{\sum_{k=1}^{M} \mathbb{I}\left[p_{\mathrm{null}}^{(k)}(i,j) < p_{\mathrm{obs}}(i,j)\right]}{M} \; , \tag{3}$$

where $\mathbb{I}(\cdot)$ is a indicator function of 1 if the specified condition is satisfied. $M$ is the total number of bootstrap ensembles. Here we have $M=5000$. Hi-C interactions with BH-FDR adjusted $p$-values $<0.01$ are chosen to be the specific interactions (Fig. 1c).

**Deep sampling of ensembles of chromatin structures using Hi-C frequencies.** We generate 3D chromatin structures under a sequential Bayesian inference framework using frequencies of specific, non-specific or all Hi-C interactions. To generate an ensemble $E$ from the Hi-C data, our goal is to maximize the probability $P(E|H)$, where $H$ represents Hi-C propensities selected as modeling features, and $E$ consists of chromatin polymers $X^{(1)}, X^{(2)}, \cdots, X^{(N)}$. By Bayes' rule[68], $P(E|H) = \frac{P(H|E)P(E)}{P(H)}$, with $P(H)$ being a constant. We generate chromatin polymers

sequentially:

$$\begin{aligned} P(E|H) &= P(E_1|H_1)P(E_2|H_2)\cdots P(E_n|H_n) \\ &= \textstyle\prod_{t=1}^{n} P(E_t|H_t) \; \propto \; \prod_{t=1}^{n} P(H_t|E_t)P(E_t) \, , \end{aligned} \tag{4}$$

where $n$ is the length of each chain, $H_t$ the selected Hi-C propensities $p_{\mathrm{obs}}(x_1, x_t)$, $p_{\mathrm{obs}}(x_2, x_t), \cdots, p_{\mathrm{obs}}(x_{t-1}, x_t)$. $E_t$ is the intermediate ensemble $X_t^{(1)}, X_t^{(2)}, \cdots, X_t^{(N)}$ at the step $t$, each chain consists of $(t-1)$ beads that are previously placed and a newly generated bead $x_t^{(k)}$.

$P(H_t|E_t)$ evaluates the similarities between simulated contact probabilities derived from chromatin conformations and Hi-C propensities. We model this term through a Poisson distribution, which is robust and can decrease the influence of the dominance of large contact counts[69]. $P(E_t)$ is reversely proportional to the number of all possible valid intermediate ensembles $E_t$ given the previously constructed $E_{t-1}$. We apply an iterative optimization strategy to find the best polymer ensemble that maximize $P(E_t|H_t)$ at each growing step (See Supplementary Methods).

**Identification of modeled single-cell domain boundaries.** We adopt a method similar to[18] to define the domain boundaries in modeled single-cell conformations. Details can be found in Supplementary Methods.

**Reporting summary.** Further information on research design is available in the Nature Research Reporting Summary linked to this article.

## Data availability

Hi-C data are downloaded from GEO database (embryos at cycles 9–13 and stages 5–8 from GSE103625, S2R+ from GSE101317). In Fig. 2, ChIP-chip datasets for clustering are downloaded from modENCODE database with IDs listed in Supplementary Table 2. In supplementary Figure 5, ChIP-chip or ChIP-seq data of H3K27me3, H3K4me3 and H3K36me3 at cycle 12 are downloaded from GSM1424916, GSM1424909 and GSM1424919. H3K27me3, H3K4me3 and H3K36me3 data at cycle 14c are downloaded from GSM1424918, GSM1424911 and GSM1424921. Expression level of gene Scyl (top) and chrb (bottom) during Drosophila embryogenesis are downloaded from the Flybase (https://flybase.org/). The source data underlying Figs. 2c–d, 3d, 4d, 5g and Supplementary Figs. 3d, 4b, 5, 7c, 7h, 8c, 9c are provided as a Source Data file. Source data are provided with this paper.

## Code availability

Source code for null model chromatin folding by fractal Monte Carlo is available via git repository at https://bitbucket.org/aperezrathke/chr-folder. Source code for Sequential Bayesian inference framework is available via https://github.com/qiusun0215/sBIF.

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

## Acknowledgements

We thank Ni Jian for assistance in figure editing and Gamze Gürsoy for useful suggestions. This work is supported by National Key Research and Development Program of China grant 2018YFC1003500, CAS-18H100000104, and NSFC-81627801 and US NIH grants R35GM127084.

## Author contributions

Q.S., Z.S. and J.L. conceived and designed the study. QS designed and implemented the sequential Bayesian inference model. A.P.R. designed and implemented the fractal Monte Carlo for null model. Q.S. carried out computation and analysis with assistance from A.P.R. D.M.C. participates in data analysis. J.L. and Z.S. supervised the overall study. Q.S. and J.L. wrote the manuscript with assistance from A.P.R., D.M.C. and Z.S. All authors read and approved the final manuscript.

## Competing interests

The authors declare no competing interests.

**Additional information**

