## [Peer Review File · Nature Communications]

Reviewers' comments:

Reviewer #1 (Remarks to the Author):

This nicely-written manuscript describes an approach for identifying significant interactions from Hi-C data and using these interactions to produce ensembles of 3D models of chromatin conformation. The idea of creating ensembles of different possible 3D conformations from Hi-C data acknowledges the fact that bulk Hi-C data reflects interactions across a population of cells and is an improvement on approaches that create single best-fit models of the Hi-C data.

While the method is novel and interesting, and should be useful for the field, the authors overstate the novelty of the biological results. I believe the manuscript would be improved by a stronger focus on comparing the method to other approaches, including quantitative comparisons and additional details about software availability and performance. This would make this method more accessible and more likely to be adopted by other members of the Hi-C community.

Major points

- The approach for identifying significant interactions from Hi-C data (here termed 'specific' interactions) is novel and sounds potentially useful, however there is no analysis of how this approach compares to other ways of identifying significant interactions. The authors should include a comparison with other methods such as GOThiC (PMID: 28379994) or Fit-Hi-C (PMID: 24501021), and perhaps also to loop-calling approaches such as HICCUPS (PMID: 25497547), given that the specific interactions identified include loop-like interactions. These should be compared both in terms of the overlap of significant interactions between different methods, and how effective these sets of interactions are for 3D modelling.
- There are multiple places where the authors overstate the novelty of their biological findings. For example, in line 283 they claim that they detect changes which would not be detectable using direct examination of Hi-C contacts, but increased chromatin compaction during *Drosophila* embryo development is found in previous studies (Hug et al. 2017 PMID: 28388407, Ogiyama et al. 2018, PMID: 30008320). Similarly, interaction dynamics at the *Scyl/chrb* locus can be detected using 4C/Hi-C data without 3D modelling (Ghavi-Helm et al. 2014 PMID: 25043061, Hug et al. 2017 PMID: 28388407, Ogiyama et al. 2018 PMID: 30008320). The establishment of boundaries at the MBT has been previously shown (Hug et al. 2017 PMID: 28388407) and the enrichment of insulator proteins at boundaries is also not novel (e.g. Hou et al. 2012 PMID: 23041285).
- When discussing the heterogeneity of the 3D conformations, the authors should be clear that this reflects theoretical heterogeneity across a population of cells, and does not necessarily reflect actual single cell conformations. Single conformations from the ensemble should not be referred to as "single cells".
- The authors only use data from S2 cells when analysing the chromatin states of regions involved in specific interactions. Given that chromatin marks are dynamic during development, their statements about the frequency of different types of interactions are not well supported. The authors should confirm whether these results are robust when using chromatin state data from another source, such as early embryos.

Minor points

- Given the nature of the chromatin fibre, one would expect to find clusters of bins involved in significant interactions with a region of interest: even if the interaction is driven by a single bin, neighbouring bins would also have elevated interaction frequency with the region of interest. This appears to be true for the *Scyl/chrb* interaction but not for *Bsg25A/slam*. Can the authors comment on why this might be the case?
- Can the authors comment on how *Scyl-chrb* interactions relate to the expression of these genes and the activity of the interacting enhancer? Are more frequent interactions associated with Polycomb repression, or with active expression?

- Line 198-200: this sentence is misleading. The fraction of significant interactions from bulk Hi-C data in a particular region does not imply anything about the fraction of interactions occurring in a single cell.
- The labelling of the samples used as "cycles" and "stages" in Fig 4 is inaccurate (cycles 9-13 also correspond to specific embryonic stages: stages 3-4) and confusing. If shorthand terms are necessary perhaps "pre-/post-MBT" would be useful terms to use.
- Line 263: No data is shown to support heterogeneous structures at the active region of interest. This should be included, at least as a supplementary figure.
- It would be nice to see single-cell analyses such as DNA FISH to support the statements about heterogeneity of single-cell chromatin conformation, particularly the idea that TAD structures are present in single cells pre-MBT. However, I realise that this may be outside the scope of this work.

Reviewer #2 (Remarks to the Author):

Review for *Nature Communications*

"High-resolution single-cell models of ensemble chromatin structures during *Drosophila* Embryogenesis from Population Hi-C" by Qiu Sun et al.

In this manuscript, the authors claim to present a new computational method that allows the reconstruction of chromatin conformations using Hi-C contacts determined in *Drosophila*. They claim that the method can identify specific functional interactions from population Hi-C maps that usually suffer from the presence of high amounts of random, non-functional chromatin contacts. Further, they argue that these specific contacts, which account for only a small fraction of all contacts, are sufficient to drive chromatin folding and reproduce topologically associated domains, long-range interactions, compaction of repressed regions, dynamic changes during embryogenesis, and more.

This paper is problematic for several reasons. The novelty appears to surround a new method, but the method is not discussed in the Abstract, Introduction, Main body, and Discussion parts of the paper. Specifically, the authors claim to *"describe a computational method for reconstruction of large 3D ensemble of single-cell chromatin conformations"*, but the paper actually does not present the method development, its validation, or its comparison with other approaches. Instead, the paper seems to mention various applications to three *Drosophila* cell types at different developmental stages.

Without a clear focus, and misleading premise, it is far from clear what the message of this paper is. If the authors want to communicate new biological insights, the focus should be on the biological interpretations. On the other hand, if they are introducing a novel method, then the method should be properly described, compared to others, validated, and so on. It appears that the method itself was submitted as a separate paper, confusing matters.

This focus problem is also exacerbated by the excessive length of the manuscript. The authors significantly exceed the allowed length for both the Abstract and main text. If the focus was clear, shortening would not be a problem, but the misleading premise and unsubstantiated claims render this manuscript premature.

In sum, a complete overhaul of the manuscript and its message would be required before this paper might be considered further, but it remains unclear to this reviewer whether a rehailed manuscript would be suitable to *Nature Communications*.

Reviewer #3 (Remarks to the Author):

The authors present a method for identifying specific Hi-C contacts that are sufficient for determining the overall 3D architecture of the chromatin fiber. They subsequently construct an ensemble of chromatin configurations consistent with the contact frequencies of the specific loci. The authors use this ensemble to extract biological insights concerning a variety of features whose heterogeneity are obscured in bulk Hi-C. Two of these – observations of three-body interactions, and observations of single-cell TAD-like structures absent in bulk Hi-C – are discussed in the comments below.

Summary: Reconstruction of configuration ensembles from bulk Hi-C is an elegant and appealing idea, and the authors convincingly show that their method qualitatively captures known features of nuclear architecture. However, I have reservations about the proposed method's use in generating quantitative conclusions, given what I perceive as a lack of bench-marking. The quantitative validity of the reconstructions is particularly important when evaluating the novel findings of three-body interactions and the appearance of TAD-like structures in individual members of the reconstructed ensemble. Additionally, very little comparison or benchmarking is given in the main text to previously reported methods in the literature which have the same goal of ensemble reconstruction. This is especially important given the authors' claims in line 55 that the community currently lacks an acceptable algorithm for ensemble reconstruction.

For these reasons, think this manuscript is acceptable for publication in Nature Communications, but only with substantial revision. The findings and methods are of interest to the community, but additional works needs to be done to clarify the novelty of the algorithm and the quantitative validity of the findings.

Specific Comments:

1. I do not believe that the null model considered here is particularly meaningful in a physical sense; why should specific interactions be those that exceed the threshold of a self-avoiding random walk (SARW) contact frequency? For example, there has been speculation that chromatin adopts a fractal globule state, which generates a different scaling of contact probability with respect to genomic distance and hence would identify different contacts as being specific interactions. The authors should provide a justification for why a SARW is a physically well-motivated choice on which to construct a null model; the current exposition is too terse in this regard.

2. A more data-driven approach to identification of specific interactions would be to take the average along each off-diagonal and then consider specific contacts to be those that have abnormally high (or low) frequencies relative to their same-distance peers. Can the authors comment on why they do not use this (in my view, more standard) approach? Including depletion of contact frequency in addition to enrichment of contact frequency might also provide useful information for the Bayesian reconstruction.

3. I would like for the authors to provide additional evidence that the ensemble constructed by the sequential Bayesian framework is in fact guaranteed to quantitatively reproduce the configurations sampled by Hi-C. Reconstruction of configurations is generally an ill-posed inverse problem with multiple solutions; for example, if the Hi-C matrix is extremely sparse, then there will be many possible configurations which satisfy the frequency constraints given by the Hi-C map. I would suggest that the authors include in-silico evidence for their claims, where it is possible to compare to ground truth; construct an ensemble, simulate Hi-C on this ensemble of configurations, and show that the proposed method is capable to reconstructing the original ensemble and not just its corresponding Hi-C map.

4. Additionally, how does the distance-adjusted Pearson correlation change as the p-value threshold for specific contacts is decreased? Pearson correlations are reported in section "3D Loop

structures recovered...” but I cannot seem to find the corresponding distance-adjusted correlations. It is also a little surprising that the all-contact maps perform worse compared to the specific-contact maps. Is this simply an issue of insufficient sampling? Can the authors provide some intuition for this?

5. It would be good to define “distance-adjusted Pearson correlation” in the Methods as opposed to just citing [37]; it’s difficult even in the cited paper to find the definition of this quantity.

6. The authors should also contrast their approach, in more detail, with existing approaches which seek to reconstruct ensembles of configurations from bulk Hi-C – for example, the approach given in main text reference [27], the approach in Supplemental Methods reference [14], the MOGEN algorithm [Trieu and Cheng, *Bioinformatics* 2016, <https://doi.org/10.1093/bioinformatics/btv754>], or the more physically motivated approach in [Zhang and Wolynes, *PNAS* 2015, <https://doi.org/10.1073/pnas.1506257112>].

7. When the sequential Bayesian inference procedure is introduced in “Specific interactions are sufficient...”, the use of “constraints” to refer to the role that specific interactions play in the reconstruction was confusing to me. In my interpretation, the role they play is to require certain contact frequencies averaged over the ensemble, but the use of the word constraints suggests that each member of the reconstruction is being constrained to have those loci in contact. My preference is that constraints be replaced by phrasing involving the word “frequency”.

In general, a high-level explanation of the method would be appreciated here, as it doesn’t appear that any is currently given; readers must refer to the Supplementary Methods in order to understand basics of the method. Given that several methods exist in the literature which purport to reconstruct ensembles of chromatin structures based on Hi-C, it would also help the reader in terms of understanding the precise way in which the proposed method is novel.

8. The authors follow changes in contacts between genomic regions identified as belonging to A (active), I (inactive), and P (polycomb). The assignment of these regions to a particular domain type is based on ChIP-seq data. Perhaps this is known, but evidence that these assignments do not change over the course of development is necessary, in my view. Otherwise, these changes in contact frequencies could be attributed to changes in domain assignment.

9. In the section demonstrating three body interactions, it would again be good to know that the sequential Bayesian framework is capable of observing changes in three-body interaction frequencies in silico, by construction of an ensemble with a particular frequency of three body interactions which is then inferred by the proposed method. The FISH comparison in Fig. S4g is a nice step, but I think it’s important that the extent of quantitative success of the method is explored in a more controlled environment.

10. Given that both the original and simulated Hi-C maps in Fig 5a seem essentially featureless, the authors’ claim to have identified TAD-like structures is surprising and interesting. This leads to two separate questions:

1. Is there perhaps a signal of TAD structures in the bulk Hi-C map? The authors can quantitatively assess this using an insulation or boundary strength score; it seems like the current evaluation is mostly qualitative.

2. To what extent is the ensemble constructed simply a reflection of random fluctuations? Said another way, if the authors were to construct an ensemble of SARWs, would they also observe a similar subpopulation of configurations with TAD-like structures? Said yet another way, can the authors provide a convincing null model here?

Revision Report

We thank all referees for their critique. The referees' thoughtful comments and suggestions are very valuable, which helped us to strengthen our paper significantly.

We have followed referees' advice closely and have revised our manuscript extensively: Much of the main text has been entirely rewritten. Our detailed responses to the referees' comments are itemized below.

Reviewer 1

Major points

1. *“(General comment) I believe the manuscript would be improved by a stronger focus on comparing the method to other approaches, including quantitative comparisons”*
“ The approach for identifying significant interactions from Hi-C data (here termed ‘specific’ interactions) is novel and sounds potentially useful, however there is no analysis of how this approach compares to other ways of identifying significant interactions. The authors should include a comparison with other methods such as GOTHiC (PMID: 28379994) or Fit-Hi-C (PMID: 24501021), and perhaps also to loop-calling approaches such as HICCUPS (PMID: 25497547), given that the specific interactions identified include loop-like interactions. These should be compared both in terms of the overlap of significant interactions between different methods, and how effective these sets of interactions are for 3D modelling.”

Thank you for these suggestions. Following your advice, we have compared our method with Fit-Hi-C, and GOTHiC. We have also examined the HICCUPS method as suggested, which is designed for loop-calling and therefore different from the other methods. We have added the following paragraphs to the **Discussion** section:

We have further compared the specific interactions identified by our method with those by Fit-Hi-C [1] and GOTHiC [2] (Supplementary Figure S4a and Supplementary Methods). Overall, all three methods identify a small fraction of specific interactions (4.1–7.5% genome-wide for S2R+ cells), with varying degree of overlaps (85.2% of Fit-Hi-C and 36.0% GOTHiC are present in ours). Compared to Fit-Hi-C and GOTHiC, our method identifies more long-range interactions (26.2% vs. 17.3% and 1.35% are of ≥ 500 kb, Supplementary Figure S4b-c). The small fractions of interactions identified by all these methods are found to be sufficient to drive chromatin folding for a 1 Mb region tested using our folding algorithm, although ours gives the highest correlation to Hi-C measurements (Supplementary Figure S4d). We also examined the effectiveness of HiCCUPS [3], which identifies looping interactions. Interestingly, we found that the much smaller fraction (0.1%) of looping interactions identified by HiCCUPS was insufficient to drive chromatin folding (Supplementary Figure S4).

and provided the following discussion:

Our method is based on an explicitly constructed random 3D polymer ensemble as the null model. This is a uniquely appropriate approach for chromatin studies, as polymer effects of random collision of chromatin fiber are explicitly modeled. This null model allowed effective discrimination of specific interactions from background noise of random chromatin collision. Several other methods such as Fit-Hi-C [1] and GOTHic [2] can also be used to identify specific interactions. However, these methods rely on null models constructed from the Hi-C data itself for removal of background noises [1, 35, 36], which may lead to inherent bias in the resulting null hypothesis test. Furthermore, our physically constructed null model can be used to identify higher-order specific many-body interactions [20], which is not possible with null models derived solely from 2D Hi-C contact maps. As shown in a separate study describing the CHROMATIX algorithm, this null model can be used for analysis of many-body chromatin interactions, such as those encountered in enhancer-rich regions [20].

We have also added more details to a new section **Comparison of specific interactions among different methods** of the **Supplementary Methods**:

We compared specific interactions identified using our method with those from Fit-Hi-C [1] (v 2.0.7), GOTHic [2] (v 1.22.0), and HICCUPS [3] (v 1.14.08). For Fit-Hi-C, we use parameters “-r 2000 -p 2”. A bias file generated by ICE is also used for the “-t” option. Fit-Hi-C-specific interactions are selected based on the criteria of q -value ≤ 0.01 . For GOTHic, we rearrange our valid Hi-C ligation products to HiCUP output format as required by GOTHicup, after removal of PCR duplicates, self-ligation and dangling reads. We perform GOTHic calculations via GOTHicup using default parameters. Hi-C contacts with q -value ≤ 0.01 are identified as GOTHic-specific interactions. For HICCUPS, we take the implementation from the juicer-tools (<https://github.com/aidenlab/juicer>) with options “--cpu -m 1024 -k KR -r 2000 -p 4 -i 10 -d 10000 -f 0.01 --ignore_sparsity”.

Enriched pixels with $FDR \leq 0.01$ are then selected. We constructed a separate 3D chromatin ensemble for each method using the respective specific interaction frequencies; all 3D ensembles were generated via sequential Bayesian inference framework. Each ensemble consisted of 50,000 single-chain chromatin polymers.

The specific interactions for a 1 Mb region (chr2L: 11–12 Mb) are shown in Figure S4a. Overall, there are 6.3% (ours), 4.1% (Fit-Hi-C), 7.5% (GOTHic) and 0.1% (HICCUPS) of genome-wide S2R+ Hi-C contacts within 4 Mb genomic distance that are identified to be specific, respectively (Supplementary Figure S4b). Despite a lower fraction retained than GOTHic (6.3% vs. 7.5%), our method identifies more long-range specific interactions (26.2% vs. 1.35% with separating distance ≥ 500 kb, respectively). The number of specific interactions

in 10^6 (M) for the whole genome is shown in the Venn diagram of Supplementary Figure S4c, along with the numbers of the intersections. Specifically, 85.2% and 36.0% of Fit-Hi-C and GOTHiC specific interactions are also identified by our method.

We further used our folding algorithm and constructed 3D ensembles of chromatin structures using contact frequencies of specific interactions obtained with these methods for the 1 Mb region of chr2L:11–12 Mb. We evaluate similarities between simulated and Hi-C contact frequencies based on Pearson correlation coefficient r and distance-adjusted correlation coefficient r' . While our folding algorithm can generate 3D chromatin ensembles using specific interactions identified by our null model, our folding algorithm can also utilize Fit-Hi-C and GOTHiC specific interactions which are also sufficient for generating 3D ensembles. However, results using our null model have the highest r and r' -values, and are most similar to the original Hi-C measurements (Supplementary Figure S4d).

We have also added Supplementary Figure S4a–4d to illustrate these new results:

Supplementary Figure S4. Comparison of specific interactions obtained using different methods. **(a)** Distributions of all and specific interactions by our method, Fit-Hi-C, GOTHiC and HICCUPS in a 1 Mb region (chr2L: 11–12 Mb). Lower left triangle shows all Hi-C contact frequencies, upper right triangle specific interactions (white and dark blue colors represent background and specific interactions). **(b)** Fractions of genome-wide specific interactions of ≤ 4 Mb distance using different methods. **(c)** Overlap of specific interactions among our method, Fit-Hi-C and GOTHiC. **(d)** Pearson correlation coefficient r and distance-adjusted correlation coefficient r' between Hi-C and simulated contact frequencies obtained from polymer ensembles constructed using specific interactions of our method, Fit-Hi-C, GOTHiC and HICCUPS.

2. “and additional details about software availability and performance.”

Source code for null model chromatin folding by fractal Monte Carlo is available via git repository at <https://bitbucket.org/aperezrathke/chr-folder>. Source code for Sequential Bayesian inference framework is available via <https://github.com/qiusun0215/sBIF>. This information appears in the **Code availability** section of the manuscript.

We have added a new figure (Supplementary Figure S8d) to provide information on computing cost in constructing ensembles of 3D polymer chains of varying lengths (100 – 1,000 beads) using frequencies of specific Hi-C contacts for the 10 regions listed in Table S1. Each ensemble consists of 50,000 polymer chains.

Supplementary Figure S8. (d) averaged CPU time (minutes) for constructing 3D polymer chains (50,000 chains, 20 Intel(R) Xeon(R) E5-2650 CPU cores) using frequencies of specific interactions for the ten regions.

3. “(General comment) the authors overstate the novelty of the biological results.”

“ For example, in line 283 they claim that they detect changes which would not be detectable using direct examination of Hi-C contacts, but increased chromatin compaction during *Drosophila* embryo development is found in previous studies (Hug et al. 2017 PMID: 28388407, Ogiyama et al. 2018, PMID: 30008320). ”

Thank you for pointing out the inaccuracy in our statements. We have modified our description, with these two very helpful references added:

Consistent with previous qualitative observations of significant changes in compactness during embryogenesis [4,5], we observed that later embryos have higher proportions of compact clusters (C4 and C5) in the polycomb-repressed region (Figure 4c). Yet our models further enable a quantification of this difference in compaction: there is an 8% reduction in the radius of gyration (R_g) [7] in this region from the pre- to post-MBT but a more significant 18% reduction in the S2R+ cells. Similarly, the end-to-end distances change by only 6% from pre- to post-MBT whereas there is an 18% change in the S2R+ cells (Supplementary Figure S9c). Similar changes in these measurements are also observed in the active region (Supplementary Figure S9d and S9i). Such quantitative measures of change in compactness can only be derived from physical models of 3D chromatin conformations.

“Similarly, interaction dynamics at the *Scyl*/*chrB* locus can be detected using 4C/Hi-C data without 3D modelling (Ghavi-Helm et al. 2014 PMID: 25043061, Hug et al. 2017 PMID: 28388407, Ogiyama et al. 2018 PMID: 30008320). ”

Thank you again for pointing out the confusing nature of our statement. We have now modified our sentences for clarification:

Previous studies suggested that the promoters of these genes and a putative enhancer physically interact and this interaction changes during embryogenesis (Figure 4e) [4, 5, 34].

To further test this idea, we constructed 3D ensembles of single-chain conformations of this region for the three cell types. As shown in Figure 4g, we found there exists a significant three-way interaction between these regions in our models. We note that the distribution of spatial distances between *chrB* and *Scyl* and two other control regions derived from our model is highly consistent with DNA FISH measurements (Fig 4d) [8]. ... Thus, our results implicate this putative enhancer [8] in forming a spatial unit of three-body interactions with the promoters of *Scyl* and *chrB*.

We also modified other relevant sentence to:

We expected that these changes in the overall compaction at the domain-level might also translate into functionally relevant changes in specific three-body interactions.

“The establishment of boundaries at the MBT has been previously shown (Hug et al. 2017 PMID: 28388407) and the enrichment of insulator proteins at boundaries is also not novel (e.g. Hou et al. 2012 PMID: 23041285).”

Indeed our writing may have been misleading. We have added references pointing to these experimental findings, and have modified our paragraph so it reads:

Previous Hi-C studies showed that, during *Drosophila* embryogenesis, TAD boundaries are only established during the MBT but are otherwise not visible on the heat map at early stages [4, 5, 9]. Indeed, as shown in Figure 2a and 4a, TADs can be clearly seen in Hi-C heat maps of post-MBT embryos (stages 5–8) and S2R+ cells, but not in early embryos.

The discussion of our results on insulator proteins and TAD boundaries has also been revised, and a citation to the important study from the Corces lab is added:

Previous studies showed that insulator proteins such as BEAF-32/CTCF/Su(Hw)/CP190 are strongly enriched at TAD boundaries of *Drosophila* [10]. We then examined whether the boundary positions appearing in our 3D models of late embryos also correlate with the preferred binding sites of these insulator proteins.

and

These results therefore show that TAD boundaries appearing in our 3D structural models of single chromatin chains, once aggregated, are consistent with findings from population-based Hi-C studies [10]. Furthermore, the heterogeneity in Hi-C data is manifested by the presence of TADs with strongly heterogeneous boundaries at an early stage, which is quantified in our modeled individual chromatin structures.

4. *“When discussing the heterogeneity of the 3D conformations, the authors should be clear that this reflects theoretical heterogeneity across a population of cells, and does not necessarily reflect actual single cell conformations. Single conformations from the ensemble should not be referred to as “single cells” ”*

Thank you for this clarifying suggestion. Following your advice, we now use the terms “modeled single-cell conformations” and “modeled single-chain” in the revised manuscript.

5. *“The authors only use data from S2 cells when analysing the chromatin states of regions involved in specific interactions. Given that chromatin marks are dynamic during development, their statements about the frequency of different types of interactions are not well supported. The authors should confirm whether these results are robust when using chromatin state data from another source, such as early embryos.”*

Thanks for pointing out this important issue. To verify whether chromatin states identified using epigenetic modifications of S2 cells are stable across different developmental stages, we have followed your advice and carried out additional computations.

We have now added the following and a new figure (Supplementary Figure S5) to **Supplementary Methods**:

To assess possible changes in the distribution of different interaction types, we use the 15 markers of S2 to categorize all specific interactions from each cell type, as the chromatin classification based on S2 cells is robust across embryos at cycle 12 and cycle 14c. This is supported by an analysis following the approach of [4] to examine the three representative histone modifications (H3K37me3, H3K36me3 and H3K4me3). Supplementary Figure S5 shows that after signals of histone modifications in 2 kb bins are aggregated over the same three chromatin states (Active, Inactive and Polycomb-repressed) defined by S2 markers, their distributions are unchanged in embryos at cycle 12 (during cycles 9–13) and cycle 14c (during stages 5–8).

Supplementary Figure S5. Distributions of H3K27me3, H3K36me3 and H3K4me3 ChIP-seq and ChIP-chip quantile-normalized signals in 2 kb bins from embryos at cycle 12, cycle 14c, and S2-DRSC are largely unchanged. The three chromatin states (Active, Inactive and Polycomb-repressed) are assigned based on S2 markers. ChIP-seq and ChIP-chip data are taken from the GEO database (Data sources listed in Supplementary Table S3). Processed wiggle files downloaded from the database are quantile-normalized before comparison.

Minor points

6. “Given the nature of the chromatin fibre, one would expect to find clusters of bins involved in significant interactions with a region of interest: even if the interaction is driven by a single bin, neighbouring bins would also have elevated interaction frequency with the region of interest. This appears to be true for the *Scyl*/*chr**b* interaction but not for *Bsg25A*/*slam*. Can the authors comment on why this might be the case?”

Thank you for raising this interesting question. We have added the following text to discuss this phenomenon (legend of Supplementary Figure S3b):

Note that bins immediately neighboring the bin containing *chr**b* also exhibit elevated interaction propensity, in contrast to that of *Bsg25A*/*slam* shown in Figure 2b. This is likely related to the much longer genomic distance of ≥ 1.6 Mb between *Bsg25A* and *slam*, compared to ~ 250 kb between *Scyl* and *chr**b* shown here. As both Figure 2b and Supplementary Figure S3b depict propensities rather than absolute frequencies, it is not apparent that the elevated propensity of *Bsg25A*-*slam* is based on read-outs of much smaller absolute frequencies prior to normalization (8) compared to that of *Scyl*-*chr**b* (37). Due to this longer genomic distance, there is overall lowered absolute frequencies of ligation between bins neighboring *slam* and the bin of *Bsg25A* and its neighbors in Hi-C experiment. Since we use a rather aggressively stringent criterion in our null model (FDR ≤ 0.01), such interactions of lower frequency with bins neighboring *slam* are not selected, in contrast to the strong elevated signal of interaction we do detect at the bin containing *slam* and its neighbors. Because of its 3D geometric basis, our method is likely more accurate in accounting for the bias due to genomic distance (1.6 Mb vs 250 kb), and therefore potentially more powerful in detecting specific long range interactions. This is evidenced from results of comparing specific interactions by our method and those by the Fit-Hi-C [1] and GOTHic [2], where our method detects more long-range (≥ 500 kb) interactions (26.2% vs. 17.3% and 1.35% for Fit-Hi-C and GOTHic, respectively).

7. “Can the authors comment on how *Scyl*-*chr**b* interactions relate to the expression of these genes and the activity of the interacting enhancer? Are more frequent interactions associated with Polycomb repression, or with active expression?”

Thank you for raising a very interesting question. We have compared our results with the expression data in the Flybase. While the full control mechanism of expression of specific genes may be quite complex, we found that *Scyl* has higher expression level at both stages 5–8 and S2R+. Overall this is consistent with the higher contact frequency of *Scyl*/enhancer interaction in later embryos we found (Supplementary Figure S9g). In addition, *chr**b* has a low expression level at stages 5–8 but its expression is elevated in S2R+. This is also consistent with the enhanced *chr**b*/enhancer contact frequency in S2R+ (Supplementary Figure S9h).

We have added a new **Supplementary Figure** summarizing what we have learned:

Supplementary Figure S6 Expression level of gene *Scyl* (top) and *chrB* (bottom) during *Drosophila* embryogenesis (data from Flybase (<https://flybase.org/>)). Both genes are expressed at all developmental stages, at least moderately. Therefore, the frequent interactions observed in our model are unlikely due to Polycomb repression. *Scyl* has higher expression level at both post-MBT (stages 5–8, blue box) and S2R+ (green box). This is overall consistent with the higher contact frequency of *Scyl*/enhancer interaction in later embryos (Supplementary Figure S9g). *chrB* has a low expression level at post-MBT but its expression is elevated in S2R+. This is consistent with the enhanced *chrB*/enhancer contact frequency in S2R+ (Supplementary Figure S9h). A full mechanistic understanding of *Scyl*/*chrB* interaction and how these two genes negotiate with the enhancer may require considerations of other possible mediators (such as architectural proteins) participating in the regulation process. These, however, are not explicitly modeled in this study.

8. *“Line 198-200: this sentence is misleading. The fraction of significant interactions from bulk Hi-C data in a particular region does not imply anything about the fraction of interactions occurring in a single cell.”*

Thank you for pointing out this issue. Indeed, the bulk Hi-C does not give direct information on individual instances of single-cells. We have now deleted this sentence.

9. *“The labelling of the samples used as “cycles” and “stages” in Fig 4 is inaccurate (cycles 9-13 also correspond to specific embryonic stages: stages 3-4) and confusing. If shorthand terms are necessary perhaps “pre-/post-MBT” would be useful terms to use.”*

We have now followed your advice and replace the labels of “cycles” and “stages” with “pre-MBT” and “post-MBT”, respectively. We have also added a sentence to the legend of Figure 4c:

pre-MBT: cycles 9–13; post-MBT: stages 5–8.

Figure 2, 4–6, Supplementary Figure S3, S6, S9–S11 have also been similarly updated.

10. *“Line 263: No data is shown to support heterogeneous structures at the active region of interest. This should be included, at least as a supplementary figure.”*

Supplementary Figure S9. (e) Hierarchical clustering of 3D chromatin configurations of all three stages of cells in the active region of chr3R: 12.8–13.9 Mb. Proportions of the 5 clusters are labeled below the aggregated heat maps. (f) Proportions of the 5 clusters in each cell type.

We have now added two new figures (Supplementary Fig S9e and S9f) to exhibit the heterogeneity of chromatin structures in the active region of chr3R: 12.8–13.0 Mb.

We have also changed a relevant sentence to:

The active region also exhibits significant heterogeneity among single-chain chromatin conformations (Supplementary Figure S9e and S9f).

11. *“It would be nice to see single-cell analyses such as DNA FISH to support the statements about heterogeneity of single-cell chromatin conformation, particularly the idea that TAD structures are present in single cells pre-MBT. However, I realise that this may be outside the scope of this work.”*

Thank you for this suggestion. Indeed, independent examination of heterogeneity of single-cell by DNA FISH data would be very helpful. As pointed out, this would be a huge undertaking requiring completely different sets of tools and algorithms. We are currently assessing the feasibility for such an in-depth analysis. Our hope is that such an analysis could be carried out in a future study.

Reviewer 2

1. (A) “*The novelty appears to surround a new method, but the method is not discussed in the Abstract, Introduction, Main body, and Discussion parts of the paper.*” “*but the paper actually does not present the method development,*”

Thank you for pointing out that we lacked certain details in our method; an unintentional oversight on our part that resulted from the word limit restrictions. We have now revised our manuscript, with the text extensively redesigned, so our method is better explained.

In the **Abstract**, we have added the following sentences:

With minimal assumptions of physical properties and without adjustable parameters, our method generates large ensembles of chromatin conformations via chain-growth deep-sampling.

In the **Results** section, we have added a new subsection of **Overview of our approach to 3D chromatin modeling**:

Our method relies on the recently developed capability of deep sampling to generate 3D chromatin ensembles [20, 22, 23]. We first identify a set of *specific chromatin interactions* from deep-sequenced population Hi-C data. They are unlikely due to ligation of randomly collided loci and are identified by comparing measured Hi-C frequencies to those simulated from an ensemble of randomly folded 3D chromatin configurations. The specific interactions identified are then used as restraints to generate an ensemble of modeled single-cell chromatin configurations for either a genomic region (200 kb to 4 Mb at 2 kb resolution) or a whole chromosome (22 Mb, at 5 kb resolution). Structural analysis, including Euclidean distance measurement and spatial clustering, is then carried out on the ensemble of modeled single-cell chromatin configurations. This provides an overall quantitative assessment of population 3D structural properties, such as compactness, radius of gyration, domain boundaries, as well as chromatin structural heterogeneity. Results from structural analysis are then integrated with other information such as enhancer/promoter information and epigenetic modifications for additional biological insight.

In the main body, we have added the following paragraphs:

Briefly, for our null model, we generate multiple chromatin chains through a fractal Monte Carlo method based on chain-growth [20, 22, 23, 31, 44], in which monomers are added one at a time. To overcome severe difficulties in sampling, this process is optimized through a recursive resampling algorithm applied at predetermined check-point lengths [20, 22, 23]. The target distribution of the ensemble is overall geometrically realizable self-avoiding chromatin chains within the specified volume confinement. This target distribution is rigorously enforced

through proper importance weighting during chain-growth (see Supplementary Methods for more details).

and

We estimate the random contact probabilities of pairs of loci by counting the frequency of 3D conformations in which the spatial distance of the corresponding pair of beads is within a distance threshold of 80 nm [16,23]. We evaluate the statistical significance of each Hi-C contact by bootstrapping the corresponding random ensemble (Figure 1b, see Methods). Hi-C interactions with BH-FDR [45] adjusted p -values below a threshold of 0.01 are then identified as specific interactions (Figure 1c and see Supplementary Methods).

and

We constructed 3D ensembles of single-chain chromatin conformations at high resolution of 2 kb using the frequencies of the specific interactions. The conformations are generated through a novel approach of sequential Bayesian inference (Figure 1d–1f, see also Supplementary Methods and Figure S1).

For each region, we construct a proper ensemble of 50,000 single-chain conformations. To constrain the ensemble according to the Hi-C interactions, we derive contact probabilities from the Hi-C frequencies. To estimate this contact probability, we take the minimalistic assumption that DNA fragments in close proximity are available for Hi-C ligations. The contact probabilities are then taken as the proportions of 3D conformations in which the spatial distances of the specific pairs of loci of interest are within the distance threshold of 80 nm. Thus, the target distribution of the chromatin ensemble is that of self-avoiding chromatin chains that satisfy the Hi-C derived contact probabilities at each chain-growing step, where beads are placed sequentially for each chain. Our sequential Bayesian inference framework ensures that the chromatin chains are consistent with the observed Hi-C, and ensemble properties can be accurately estimated (Supplementary Figure S7). We then aggregate single-chain conformations to obtain the simulated Hi-C contact maps (Figure 3a, Supplementary Figure S8a).

In the **Discussion** section, we have added the following text:

The computational method described in this study can quantitatively connect statistical patterns in Hi-C maps to physical 3D chromatin structures.

Furthermore, our method can bridge the gap between high-resolution population Hi-C studies and fine-detailed single-cell 3D structures of chromatin [6, 37–43],

We have also provided many more details of our method, including a figure detailing the computational time. Due to the space limitation, they appear in the **Supplementary Methods** section.

(B) “its comparison with other approaches.”

To address this concern, we have revised our manuscript extensively and provide more specifics to clarify this important issue. We have added to the **Discussion** section the following to compare our approach with those of others:

Our method is based on an explicitly constructed random 3D polymer ensemble as the null model. This is a uniquely appropriate approach for chromatin studies, as polymer effects of random collision of chromatin fiber are explicitly modeled. This null model allowed effective discrimination of specific interactions from background noise of random chromatin collision. Several other methods such as Fit-Hi-C [1] and GOTHic [2] can also be used to identify specific interactions. However, these methods rely on null models constructed from the Hi-C data itself for removal of background noises [1, 35, 36], which may lead to inherent bias in the resulting null hypothesis test. Furthermore, our physically constructed null model can be used to identify higher-order specific many-body interactions [20], which is not possible with null models derived solely from 2D Hi-C contact maps.

and

Our approach for constructing 3D ensembles differs from several existing methods [16–19, 26]. With only basic physical considerations of fiber density and ligation distance threshold, it is minimalistic and there are no adjustable parameters. No chromatin states are assigned to polymer beads, and there are no *a priori* assumptions on locations of loop anchors. Furthermore, our method samples well. With deep sampling enabled through Bayesian sequential inference, our method is unique in its ability to generate a large number (e.g. 5×10^4) of diverse chromatin conformations that are consistent with Hi-C measurements.

The following paragraphs has also been added to **Discussion** to describe the advantages of our method in characterizing chromatin heterogeneity.

With the ability to convert 2D high-resolution population Hi-C heat maps into $\sim 5.0 \times 10^4$ modeled single-cell chromatin conformations at high resolution, our method can provide quantification of chromatin heterogeneity (Fig 4c). This will allow possible differentiation of chromatin configurations that are functional from those that are not.

We have also provided further detailed discussion. Following your advice of limiting the length of the article, these are now described in the **Supplementary Methods**:

Our method. Our method differs from several other methods [16–19, 26] in important regards:

Minimal assumptions. Our model is minimalistic. It does not require assigning each bead to a specific chromosome state. There is also no *a priori*

assumption on where loop anchor sites are located (e.g. CTCF). Rather, it requires only a small number of contact probabilities at different loci pairs. These are the specific interactions derived from Hi-C data analysis using our null model. With basic physical assumptions (chromatin fiber density, self-avoiding chains, and Euclidean distance threshold in nm for ligation), there are no adjustable model parameters. The target distribution from which our Hi-C ensembles are constructed is that of self-avoiding chromatin chains satisfying Hi-C derived pairing probabilities.

This minimalistic approach offers a useful advantage, namely, it avoids another level of complexity in deconvolving degeneracy due to a large parameter space, which is in addition to the complexity of deconvolving Hi-C heat maps into an ensemble of 3D chromatin configurations. In models where chromatin beads are assigned to different states and values of binding affinities are required, there are likely many permutations of states and binding affinities that are all consistent with the Hi-C data. Our model avoids this potential problem of limited identifiability.

Deep sampling. Our method also differs from several existing methods in an important aspect. We make use of a well-designed sampling distribution to improve sampling efficiency, which is dynamically adjusted based on the chromatin polymer ensemble at each growing step (Supplementary Figure S2a). The sampled ensemble rigorously follows that of the predefined target distribution, *i.e.*, self-avoiding chromatin chains satisfying Hi-C derived pairing probabilities. This in contrast to several optimization-based method (such as the method of [17]), where each chain is optimized separately through an objective function constructed on single chains (Supplementary Figure S2b).

Our sampling method is built upon significant advancements in sequential importance sampling techniques [20, 30]. As a result, we achieve very deep (*i.e.*, large-scale) sampling. For example, the ensembles we constructed contain 50,000 chromatin configurations, each of 1,000 beads of 2 kb (for loci) or 4,485 beads of 5 kb (for chromosome X), and are independently grown by sequentially adding one bead at a time. With advances in resampling and rejection control techniques far beyond the well-known PERM approach [32], we are able to overcome the well-recognized attrition problem of severe difficulty in growing long SAW chains in confined volume [22, 30]. Our sampling method is also fast. This is shown in Supplementary Figure S8d, where the averaged CPU costs (in minutes) for construction of ensembles for ten regions by specific interactions at different chain length (100–1,000 beads) are provided.

This level of deep sampling likely compares very favorably with previous studies. Under ergodic assumptions, both molecular dynamics (MD) simulations and Markov chain Monte Carlo can generate proper samples after long runs and establishment of convergence. However, achieving convergence is nontrivial, and obtaining decorrelated samples at accessible time scales along simulation trajectories is challenging in both cases. Our ensembles of 5×10^4 chromatin

polymers from chain growth are results of better sampling. Compared to MD simulation, our method does not suffer from the issue of in-sufficient simulation time. Compared to MCMC simulations, our method does not suffer from possible slow convergence, which is currently an unsolved problem. In general, our method can generate more quality samples in a shorter time frame, therefore is better equipped at quantifying heterogeneity than those from 2×10^4 steps of MD simulations reported in [26], the 5,000 conformations generated using Metropolis Monte Carlo reported in [16], or typically a few hundred structures obtained using methods based on optimization [18].

Realistic ligation model. Our probabilistic model of ligation is realistic with less assumptions. We assume regions in close spatial proximity below a threshold are available to form Hi-C ligations probabilistically. Hi-C probabilities therefore represents the proportions of polymer chains that satisfy the ligation threshold (Supplementary Figure S2c). In contrast to the approach of [17], where Hi-C contact frequencies are translated to spatial distances via a reverse exponential function under implicit assumptions regarding the rate of chromatin diffusion and how it relates to the mean contact frequency, our model makes no such assumptions (Supplementary Figure S2d).

High resolution. The resolution of our model is high: 2 kb for specific loci, and 5 kb for whole chromosome, which is at the same resolution of many Hi-C studies.

Overall, our method is well-suited to a number of tasks, such as quantitative assessment of chromatin heterogeneity and determination of three-body interactions. It enabled us to reveal the novel insights reported in this study.

These statements are made based on comparison with four influential methods, which are representative of the two prominent classes of approaches, namely, physically-based and data-driven methods for Hi-C 3D structures modeling [33].

Furthermore, we have added a new section of **Comparison with Other Methods in 3D modeling** in **Supplementary Methods** to elaborate on the differences between ours and these methods. We first discuss these methods, then point out how our method differs. The first section on physically-based methods reads:

Monte Carlo model of ref [16]. In [16], an artificial square-well potential was employed, with the depth, width and location of the potential optimized through curve-fitting to the 5C-data modeled. This optimized potential was then used to generate 5,000 conformations through Metropolis Monte Carlo.

Validation was in the form of agreement with the distributions, mean, and standard deviation of pair-distances of genomic regions measured by FISH. Similar validation is provided in our study (Fig 4d), where we showed our model can reproduce the distance distributions between Scyl-chrb and distributions of two other pair-interactions. As this study is based on 5C data, no results on comparing simulation with Hi-C data were reported.

MiChroM model. The MiChroM method [19, 26] investigates the energy landscape of chromatin folding. Its aim is to study general folding properties emerging from different factors in the chromatin energy model. A fully specified energy model requires A) additional information along the chromosome. Specifically, A1) beads are assigned to a specific chromatin states; and A2) the locations of loop anchors are specified (such as CTCF sites). Further, there are B) 27 parameters for the physical interactions: interaction energy of beads of different chromatin states; interaction energy of loop anchors; and distance-dependent compaction.

Both A1) and A2) are obtained based on annotation provided by Rao et al [3]. The 27 adjustable parameters are estimated from Hi-C data of chromosome 10. The model is then applied to other chromosomes, in which 2×10^4 steps of MD simulations were carried out to generate chromatin ensembles. Overall, Hi-C contact maps can be reproduced at r of 0.956. A number of important findings were made with this model, including the likely origin of chromosome territories, phase separation among chromatin types, active genes in the periphery of chromosome territories, and knottedness of chromatin (see also a commentary on this work [25]).

As it was designed to study whole genome folding, the resolution of MiChroM is necessarily limited to 50 kb. Therefore, this model does not provide detailed spatial structures below 50 kb. At a whole chromosome level, our method has similar Pearson correlation coefficient ($r = 0.94$ for *Drosophila* chr X), but at a higher resolution of 5 kb.

We have also added discussion on the MOGEN method, a well-known representative of the data driven methods. This is a method based on optimization. The objective function is empirically constructed to enforce contacts and reduce clashes, although physical interpretations appear to be more difficult compared to MiChroM. The outcome of such optimization are connected coordinates that form a chromatin chain, with Hi-C information encoded in the objective function satisfied probabilistically through optimization.

The authors reported that 500 whole-genome chromosome structures were obtained using Hi-C data, likely at the same resolution of 1 Mb as reported for a synthetic data set. Reported results were of a coarse-grained nature, e.g., mean distances of centers of mass (CoM) of individual chromosomes to CoM of the whole genome. No detailed results beyond these coarse-grained findings were provided, and no correlation between simulated and measured Hi-C contact maps were given.

We have added the following discussion on MOGEN to the SI:

MOGEN. The MOGEN method is data-driven and is based on optimization. It employs an empirical objective function designed to satisfy as many high probability restraints as possible, while reducing clashes. Restraints are converted from Hi-C data based on frequency thresholds. Parameters are adjusted so chains can be generated successfully after optimization by gradient ascent. MOGEN was

used to generate a set of 500 genomewide chromosome structures using Hi-C data. However, it is unclear how well this approach works at finer base-pair resolutions (e.g. 5 kb).

Furthermore, we have also consulted the method of Wang et al [17]. This is another data-driven method based on 3C data in which each chromatin chain assumes an idealized energy (excluded volume, bending, and stretching). This energy model and 3C measurements are then combined under a Bayesian framework. The Expectation-Maximization algorithm is employed to infer unknown parameters so the likelihood function can be calculated, and the posterior probability distribution of the chromatin conformations given the 3C data can be inferred.

The energy model appears to be less well motivated than MiChroM, as bending and stretching terms do not appear to be well justified physically. Validation of this method was based on FISH studies of the yeast genome, in which modeled distances are compared with FISH measurements. Similar validation is provided in our study (Figure 4d), where we showed that our model can reproduce the distance distributions between Scyl-chrb and distributions of two other pair-interactions.

As simulated contact maps were not reported in this paper, direct comparison between this 3C-based method and our Hi-C models (at 2 kb or 5 kb resolution) is not possible. In fact, more comprehensive studies and predictions on yeast genome structures were carried out in our previous study [27], where we compared our simulated Hi-C maps with measured Hi-C map. Both inter- and intra-chromosomal Hi-C contacts can be predicted ($r = 0.82$ and $r = 0.91$, respectively). We believe our genome-wide Hi-C based models are strong results.

We have added the following to the SI:

3C-based Model. The method of Wang et al [17] was developed for analysis of 3C data, in which chromosomes are modeled as connected fragments, whose lengths are determined by restriction enzyme cuts. The idealized energy (excluded volume, bending, and stretching) of each chromatin chain is combined with 3C measurements under a Bayesian framework, where parameters of a Gaussian-based likelihood function is inferred using the EM algorithm. This method is used to generate an ensemble of yeast genome structures, with the results validated by comparing distance distributions with FISH measurements.

While chromatin structures reported are at fragment resolution, there are other studies of budding yeast, where simulated and measured Hi-C heat maps show excellent correlations [17, 27, 28]. For example, our earlier ensemble models of yeast genome based on a random model similar to that of this study exhibit an excellent whole-genome correlation ($r = 0.82$) at 15 kb resolution [27].

To compare with other methods in identifying specific interactions, we have compared our methods with the methods of Fit-Hi-C, GOTHIC, and HICCUPS. We have added the following to **Discussion**:

We have further compared the specific interactions identified by our method with those by Fit-Hi-C [1] and GOTHic [2] (Supplementary Figure S4a and Supplementary Methods). Overall, all three methods identify a small fraction of specific interactions (4.1–7.5% genome-wide for S2R+ cells), with varying degree of overlaps (85.2% of Fit-Hi-C and 36.0% GOTHic are present in ours). Compared to Fit-Hi-C and GOTHic, our method identifies more long-range interactions (26.2% vs. 17.3% and 1.35% are of ≥ 500 kb, Supplementary Figure S4b-c). The small fractions of interactions identified by all these methods are found to be sufficient to drive chromatin folding for a 1 Mb region tested using our folding algorithm, although ours gives the highest correlation to Hi-C measurements (Supplementary Figure S4d). We also examined the effectiveness of HiCCUPS [3], which identifies looping interactions. Interestingly, we found that the much smaller fraction (0.1%) of looping interactions identified by HiCCUPS was insufficient to drive chromatin folding (Supplementary Figure S4).

(B) “its validation”,

As validation of our method, we have presented the following results in the main text:

- (a) Successful reproduction of population Hi-C heat-maps for 10 regions and for the whole X chromosome, with excellent correlations ($r = 0.91$ – 0.98 and $r = 0.94$, see Figure 3 and the section **Specific interactions are sufficient to drive chromatin folding in *Drosophila***);
- (b) Establishment of the concordance of Euclidean distance measurement from our model and from FISH experimental data (Figure 4d), where we showed that our model can reproduce the distance distributions between Scyl-chrb and distributions of two other pair-interactions measured by FISH;
- (c) The clarity of biological patterns such as the stage-specific increase/decrease in certain interaction types (Figure 2c and 2d) emerged from our model. This provides another indication of the validity of our method, as no such patterns can be observed in the unprocessed Hi-C data.

To further demonstrate the extent of quantitative success of our method in a more controlled environment, we have carried out additional simulations using a system suggested by Reviewer 3. We have added our results to a new section **In silico evaluation of the sequential Bayesian inference framework** in **Supplementary Methods**:

To quantitatively evaluate the capability in reconstructing chromatin configurations based on Hi-C measurements of our method, we first construct an original polymer ensemble as the “ground truth”. This original ensemble consists of 50,000 polymer chains, each of 200 beads. To simulate domain structures, we added an intra-TAD contact frequency of 0.3 to each pair of nodes localized between 21 and 60, between 61 and 120, and between 121 and 180. Here contacts are defined as ≤ 80 nm Euclidean distance. We added another inter-TAD

contact frequency of 0.1 to each pair of nodes, with one node localized between 61 and 120 and the other between 121 and 180. We also added a loop contact frequency of 0.8 to one loop interaction, with anchors p_1 localized in 59 – 63 and p_2 in 119 – 123, and another loop interaction with anchors p_1 localized in 59 – 63 and p_3 in 179 – 183. These contact frequencies are artificially imposed and are used to provide “driving forces” for TAD formation and loop formation.

We then aggregate the contact maps of each single chromatin polymer in our original ensemble and obtain our simulated target “Hi-C” map, from which we then reconstruct another polymer ensemble via our sequential Bayesian inference framework. This reconstructed ensemble consists of 50,000 chromatin polymer chains (Supplementary Figure S7a).

We find beads exhibiting similar distance distributions to other beads, when corresponding beads in the two ensembles are compared. This is demonstrated in Supplementary Figure S7b, which depicts the distance distributions of 5 selected beads (0-, 40-, 80-, 120-, and 160-th) to all other beads in box plots for the two ensembles. Furthermore, the median fractions of bead pairs between these selected beads and all other beads that are within 80nm are also similar: 4.2% vs. 4.3% for the 0-th node (original vs. reconstructed), 11.9% vs. 12.2% for the 40-th node, 14.9% vs. 14.3% for the 80-th node, 15.5% vs. 15.8% for the 120-th node and 14.2% vs. 13.8% for the 160-th node. We further computed the radius of gyration and end-to-end distance for each simulated polymer chain in both ensembles, and find that there is no statistical difference between the original and the reconstructed ensemble (Supplementary Figure S7c).

We then combine the original and the reconstructed ensembles by randomly selecting 5,000 chains from each ensemble. We further group the mixtures of polymer chains into clusters using hierarchical clustering. This results in 7 structural clusters (Supplementary Figure S7d) judged by the silhouette scores (Supplementary Figure S7e). For these 7 clusters, the proportion of each cluster is highly consistent between the original and the reconstructed ensemble (Supplementary Figure S7f). Furthermore, the aggregated contact heat maps of polymer chains from the original ensemble are also strongly similar to those from the reconstructed ensemble for all 7 clusters (Supplementary Figure S7g). In addition, the reconstructed contact map of the full ensemble is also strongly similar to the original map ($r = 0.98$, $r' = 0.86$).

Furthermore, we have examined the spatial distance distributions among three anchors p_1 , p_2 , and p_3 in a simulated region containing a unit of three-body interactions. The distribution of distances p_1-p_2 , p_1-p_3 and p_2-p_3 are overall very similar between the original and the reconstructed ensembles (see Supplementary Figure S7h shown below). Moreover, the fractions of the loop pairs and the three-body contacts $p_1-p_2-p_3$ are highly consistent between those from the original and those from the reconstructed ensembles (Supplementary Figure S7i).

Supplementary Figure S7. An illustration of in-silico simulation procedure for model verification. **(a)** First, we construct an original chromatin polymer ensemble and aggregate the single-chain contact maps into a simulated Hi-C contact map; we then apply our sequential Bayesian inference framework and reconstruct another polymer ensemble solely from the simulated Hi-C contact map. **(b)** Distance distributions of 5 specific nodes (0-th, 40-th, 80-th, 120-th and 160-th) with all other nodes (less than 80 nm threshold). **(c)** Distributions of radius of gyration (R_g) and End-to-End distance of the original and the reconstructed ensemble. Wilcoxon rank sum test is used for p -value calculation. **(d)** Hierarchical clustering of chromatin polymer chains after combining the original and reconstructed ensemble (randomly select 5,000 chains from each ensemble for clustering). **(e)** Silhouette scores for different choices of cluster numbers. **(f)** Distributions of cluster proportions of the original and the reconstructed ensemble. **(g)** Pearson correlation coefficient r and distance-adjusted correlation coefficient r' between aggregated contact maps from the original and the reconstructed ensemble for each cluster.

We have added the following sentence to **Supplementary Methods**:

Our approach can uncover 3-body interaction unit, as shown in Supplementary Figure S7h and S7i.

Supplementary Figure S7. (h) Distance distributions of p1-p2, p1-p3, and p2-p3 looping contact pairs labeled in (a) for the original and reconstructed ensembles. (i) Fractions of p1-p2, p1-p3, p2-p3 looping pairs and p1-p2-p3 three-body contact pairs in the original and reconstructed ensembles.

2. *“it is far from clear what the message of this paper is. If the authors want to communicate new biological insights, the focus should be on the biological interpretations. On the other hand, if they are introducing a novel method, then the method should be properly described, compared to others, validated, and so on.”*

Thank you for pointing out the confusion in our writing. Our goal is to introduce a computational method and to illustrate important principles enabled by this method, such as the role of driver interactions in chromatin folding, the nature of single-cell TADs and its relationship with population TADs, and developmental-stage dependent changes in chromatin structures. Providing analysis using real biological data is necessary, so we can demonstrate these and other specific interesting biological findings. Such demonstrations in turn can motivate others to apply this new method in their studies. Results from these studies would provide useful insight, which may in turn help to formulate new hypotheses and new design experiments.

We have rewritten/reconstructed our manuscript extensively, so this logic is better represented. In the **Introduction** section, we now point out several important challenging issues in chromatin studies, and then point out how our work can help to resolve these issues. In the **Results** section, we now provide: 1) a general description of the new method enabling the biological findings, with focus on key aspects of the method and discussion on their implications in understanding 3D chromatin structures. This method description helps readers to make sense of the biological findings. 2) illustrative examples on the type of biological questions that now can be answered using our method. Overall, we believe we have improved the clarity significantly.

Following your advice of controlling article length, more technical aspects of our work are now placed in the **Supplementary Methods**.

“It appears that the method itself was submitted as a separate paper, confusing matters.”

This is our oversight. This additional manuscript (now published as [20]) was provided to reviewers to offer information on how well our overall approach works in constructing high quality models of 3D chromatin structures.

This separate manuscript describes a different biological concept/problem, namely, the characterization of the functional landscape of multivalent chromatin interactions in transcriptionally active loci. The biological system is that of mammalian GM122878 cells instead of *Drosophila* [20]. Furthermore, the CHROMATIX method described in that paper is designed differently and solves a different computational problem.

Specifically, the CHROMATIX method is about detection of many-body interactions that are specific. This is done with a different constrained folding algorithm, in which a distribution of directed acyclic graphs (DAG) encoding possible causal relationships among contacts is used [20]. Although the null model is the same and certain aspects of sampling are related, key aspects of the algorithms are different from the current study.

We have added the following sentences to the manuscript for clarification:

As shown in a separate study describing the CHROMATIX algorithm, this null model can be used for analysis of many-body chromatin interactions, such as those encountered in enhancer-rich regions [20].

3. *“This focus problem is also exacerbated by the excessive length of the manuscript. The authors significantly exceed the allowed length for both the Abstract and main text. If the focus was clear, shortening would not be a problem, but the misleading premise and unsubstantiated claims render this manuscript premature.”*

The reviewer’s point is well-taken. The abstract and the main text are now in compliance, as per consultation with the editor Dr. Carolina Perdigoto at *Nat Comm* on length limits. We will continue to work closely with Dr. Perdigoto to ensure our manuscript is not over the length limit of the journal.

“In sum, a complete overhaul of the manuscript and its message would be required before this paper might be considered further”

We have followed your advice and have extensively revised all important sections of the manuscript, so the focus and logic of our study is now presented clearly. In addition, extensive information is now provided for validation of our method, along with comparison with other methods. We believe this overhauled new version has sharpened our message and the issues raised have been clarified for the general audience of *Nat Comm*.

Reviewer 3

1. *“I do not believe that the null model considered here is particularly meaningful in a physical sense; why should specific interactions be those that exceed the threshold of a self-avoiding random walk (SARW) contact frequency? For example, there has been speculation that chromatin adopts a fractal globule state, which generates a different scaling of contact probability with respect to genomic distance and hence would identify different contacts as being specific interactions. The authors should provide a justification for why a SARW is a physically well-motivated choice on which to construct a null model; the current exposition is too terse in this regard.”*

Thank you for this suggestion. We have added a paragraph to justify our approach in the **Supplementary Methods**:

Our null model is based on the considerations that the excluded volume of self-avoiding chromatin chains and the confinement of the cell nucleus impose strong restrictions on the available space for chromatin. As experimental treatments can cross-link genomic elements within certain spatial distances, regardless of whether specific interactions exist, many Hi-C detected frequencies may be due to such random non-specific bystander collision contacts [21]. Our null model is designed to identify Hi-C contact pairs which cannot be accounted for by such random collision due to the physical properties of excluded volume, chain connectivity, and volume confinement. The specific contact pairs are selected rather conservatively following a stringent p -value and FDR criterion.

The issue of scaling relationship is very important and was the subject of an earlier study of ours published in 2014 [22], where we demonstrated that our random polymer model of self-avoiding walks under volume confinement are sufficient to give rise to the experimentally observed scaling-relationships (contour-length and looping probability vs. genomic distances), which were the experimental observations motivated the original fractal globule model. In addition, our model can also account for the plateau effect that is not well explained by the fractal globule model. Furthermore, there is no tunable parameters in our model. The effectiveness of our random polymer model was further demonstrated in a subsequent study [27], where we showed that with a few additional biological landmarks as restraints following [28,29], we can reproduce the genome-wide Hi-C heat map of budding yeast at 15 kb resolution, including both intra-chromosome and inter-chromosome contacts by improving upon earlier studies of [28,29].

We have added the following sentences in **Supplementary Methods** for clarification:

Previous studies showed that this random polymer model of self-avoiding connected polymers with excluded volume can reproduce experimentally observed scaling relationships of contour-length vs genomic distance (as measured by FISH) and looping probability vs genomic distance (as measured by Hi-C) [22]. With a few additional biological landmarks, it has also been successful in reproducing genome-wide Hi-C contact maps of budding yeast [27].

Our null model has another advantage, which is now included in **Discussion**:

Furthermore, our physically constructed null model can be used to identify higher-order specific many-body interactions [20], which is not possible with null models derived solely from 2D Hi-C contact maps. As shown in a separate study describing the CHROMATIX algorithm, this null model can be used for analysis of many-body chromatin interactions, such as those encountered in enhancer-rich regions [20].

It is also interesting to note that from a subsequent study published in 2015 [24], Dr. Lieberman-Aiden, who was the lead author of the original Hi-C study where the fractal-globule model was proposed, seems to have rescinded this idea: “our measurements of $\gamma \approx 0.75$ inside contact domains are inconsistent with the hypothesis that contact domains tend to form fractal globules”, “Physical simulations suggest that $\gamma \approx 0.75$ is consistent with an unknotted, nonequilibrium state that is anisotropic rather than fractal.” ([24], p. E6465).

2. “A more data-driven approach to identification of specific interactions would be to take the average along each off-diagonal and then consider specific contacts to be those that have abnormally high (or low) frequencies relative to their same-distance peers. Can the authors comment on why they do not use this (in my view, more standard) approach?”

Thank you for pointing out this interesting approach for identifying specific interactions. Respectfully, we believe our method is more direct and more physically motivated. It is also more effective in several important aspects. We have added the following paragraphs in the section **Null model construction via Fractal Monte Carlo** in the **Supplementary Methods** to elaborate on this:

An intuitive data-driven approach for identifying specific Hi-C interactions is to take the average along each off-diagonal and then consider specific contacts to be those that have abnormally high (or low) frequencies relative to their same-distance peers. This data-driven approach has some disadvantages. First, it uses the measured Hi-C to estimate a background model serving as a null distribution. This can bias the calculation of statistical significance of Hi-C contacts, as the specific contacts to be detected themselves are integral to the generation of the null distribution. Second, as there are less and less contact pairs for longer-range off-diagonals, detecting specific long range contact pairs becomes difficult without resorting to certain *ad-hoc* interpolation/extrapolation across neighboring diagonals. Our method avoids this difficulty, as it has a completely decoupled generative random model, which can generate a sufficiently large number of samples, allowing reliably detection of long-range interactions.

“Including depletion of contact frequency in addition to enrichment of contact frequency might also provide useful information for the Bayesian reconstruction.”

Thanks for this advice. Indeed, incorporating depletion of contact frequency into a Bayesian inference framework will be very helpful. At the current stage, our rather conservative approach of considering only enriched Hi-C contacts appears to be adequate, as our ensembles do exhibit strong similarity to the original Hi-C data. We will further assess and consider implement this idea in the future.

3. *“(General comment) However, I have reservations about the proposed method’s use in generating quantitative conclusions, given what I perceive as a lack of bench-marking.”*
“I would like for the authors to provide additional evidence that the ensemble constructed by the sequential Bayesian framework is in fact guaranteed to quantitatively reproduce the configurations sampled by Hi-C. Reconstruction of configurations is generally an ill-posed inverse problem with multiple solutions; for example, if the Hi-C matrix is extremely sparse, then there will be many possible configurations which satisfy the frequency constraints given by the Hi-C map. I would suggest that the authors include in-silico evidence for their claims, where it is possible to compare to ground truth; construct an ensemble, simulate Hi-C on this ensemble of configurations, and show that the proposed method is capable to reconstructing the original ensemble and not just its corresponding Hi-C map.”

Thank you for this very helpful suggestion and indeed it is challenging to solve the inverse problem of reconstructing chromatin ensembles. We have followed your advice and carried out additional *in silico* studies to examine whether our predicted ensemble are structurally consistent with the specifications of the original ensemble. We have added our results to a new section **In silico evaluation of the sequential Bayesian inference framework** in **Supplementary Methods**:

To quantitatively evaluate the capability in reconstructing chromatin configurations based on Hi-C measurements of our method, we first construct an original polymer ensemble as the “ground truth”. This original ensemble consists of 50,000 polymer chains, each of 200 beads. To simulate domain structures, we added an intra-TAD contact frequency of 0.3 to each pair of nodes localized between 21 and 60, between 61 and 120, and between 121 and 180. Here contacts are defined as ≤ 80 nm Euclidean distance. We added another inter-TAD contact frequency of 0.1 to each pair of nodes, with one node localized between 61 and 120 and the other between 121 and 180. We also added a loop contact frequency of 0.8 to one loop interaction, with anchors p1 localized in 59 – 63 and p2 in 119 – 123, and another loop interaction with anchors p1 localized in 59 – 63 and p3 in 179 – 183. These contact frequencies are artificially imposed and are used to provide “driving forces” for TAD formation and loop formation.

We then aggregate the contact maps of each single chromatin polymer in our original ensemble and obtain our simulated target “Hi-C” map, from which we then reconstruct another polymer ensemble via our sequential Bayesian inference

framework. This reconstructed ensemble consists of 50,000 chromatin polymer chains (Supplementary Figure S7a).

We find beads exhibiting similar distance distributions to other beads, when corresponding beads in the two ensembles are compared. This is demonstrated in Supplementary Figure S7b, which depicts the distance distributions of 5 selected beads (0-, 40-, 80-, 120-, and 160-th) to all other beads in box plots for the two ensembles. Furthermore, the median fractions of bead pairs between these selected beads and all other beads that are within 80nm are also similar: 4.2% vs. 4.3% for the 0-th node (original vs. reconstructed), 11.9% vs. 12.2% for the 40-th node, 14.9% vs. 14.3% for the 80-th node, 15.5% vs. 15.8% for the 120-th node and 14.2% vs. 13.8% for the 160-th node. We further computed the radius of gyration and end-to-end distance for each simulated polymer chain in both ensembles, and find that there is no statistical difference between the original and the reconstructed ensemble (Supplementary Figure S7c).

We then combine the original and the reconstructed ensembles by randomly selecting 5,000 chains from each ensemble. We further group the mixtures of polymer chains into clusters using hierarchical clustering. This results in 7 structural clusters (Supplementary Figure S7d) judged by the silhouette scores (Supplementary Figure S7e). For these 7 clusters, the proportion of each cluster is highly consistent between the original and the reconstructed ensemble (Supplementary Figure S7f). Furthermore, the aggregated contact heat maps of polymer chains from the original ensemble are also strongly similar to those from the reconstructed ensemble for all 7 clusters (Supplementary Figure S7g). In addition, the reconstructed contact map of the full ensemble is also strongly similar to the original map ($r = 0.98$, $r' = 0.86$).

Supplementary Figure S7. An illustration of in-silico simulation procedure for model verification. **(a)** First, we construct an original chromatin polymer ensemble and aggregate the single-chain contact maps into a simulated Hi-C contact map; we then apply our sequential Bayesian inference framework and reconstruct another polymer ensemble solely from the simulated Hi-C contact map. **(b)** Distance distributions of 5 specific nodes (0-th, 40-th, 80-th, 120-th and 160-th) with all other nodes (less than 80 nm threshold). **(c)** Distributions of radius of gyration (R_g) and End-to-End distance of the original and the reconstructed ensemble. Wilcoxon rank sum test is used for p -value calculation. **(d)** Hierarchical clustering of chromatin polymer chains after combining the original and reconstructed ensemble (randomly select 5,000 chains from each ensemble for clustering). **(e)** Silhouette scores for different choices of cluster numbers. **(f)** Distributions of cluster proportions of the original and the reconstructed ensemble. **(g)** Pearson correlation coefficient r and distance-adjusted correlation coefficient r' between aggregated contact maps from the original and the reconstructed ensemble for each cluster.

4. “(1) Additionally, how does the distance-adjusted Pearson correlation change as the p -value threshold for specific contacts is decreased?”

We have investigated this issue following your advice. Overall, the changes are rather small. We have added a new figure (Supplementary Figure S8e), where both the distance-adjusted (r') and the original (r) Pearson correlation coefficients are shown when different p -value thresholds are used for chr2L 11.0–12.0 Mb.

Supplementary Figure S8. (e) Pearson correlation coefficient (r) and distance-adjusted correlation coefficient (r') between Hi-C and simulated contact maps under different thresholds of BH-FDR adjusted p -value for the 1 Mb region of chr2L: 11.0–12.0 Mb.

“(2) Pearson correlations are reported in section “3D Loop structures recovered...” but I cannot seem to find the corresponding distance-adjusted correlations.”

We apologize for this oversight. We have updated our **Results** section so the correlations are now provided:

Simulated heat maps of contact probability using all and specific interactions exhibit strong similarities to the corresponding heat maps of measured Hi-C ($r = 0.92 - 0.98$, $r' = 0.56 - 0.81$ and $r = 0.91 - 0.98$, $r' = 0.58 - 0.81$, respectively, and see Supplementary Table S1). In contrast, simulated Hi-C heat maps using nonspecific interactions fail to capture much of the structural features observed in Hi-C maps ($r = 0.48 - 0.58$, $r' = -0.02 - 0.27$, Figure 3a and 3c).

We have also provided additional information on correlations in Supplementary Table S1.

“(3) It is also a little surprising that the all-contact maps perform worse compared to the specific-contact maps. Is this simply an issue of insufficient sampling? Can the authors provide some intuition for this?”

Thank you for raising this question. Details of r' and r values for all 10 loci showed that structural ensembles built by specific and all interactions are overall comparable, although among the 10 ensembles, the poorest r for specific contacts ($r = 0.92$) are better than that from all contacts ($r = 0.91$).

We have modified our sentence to avoid any confusion:

Ensembles generated using all Hi-C frequencies and using only specific interactions have similar correlations (see Supplementary Table S1). However, the latter can recover structural features such as loops with better clarity (Figure 3a).

5. *“It would be good to define “distance-adjusted Pearson correlation” in the Methods as opposed to just citing [37]; it’s difficult even in the cited paper to find the definition of this quantity.”*

We have added the following sentences in a new section **Calculation of distance-adjusted Pearson correlation coefficient** in **Supplementary Methods**:

To eliminate the effects of genomic distance, we measure the distance-adjusted Pearson correlation r' following ref [14]. Specifically, for each diagonal of the contact matrices, we calculate and subtract the averaged contact frequency at that specific genomic distance for both Hi-C and simulated contact maps. We then calculate the Pearson correlation coefficient after subtraction.

6. *“(General comment) Additionally, very little comparison or benchmarking is given in the main text to previously reported methods in the literature which have the same goal of ensemble reconstruction.”, “The authors should also contrast their approach, in more detail, with existing approaches which seek to reconstruct ensembles of configurations from bulk Hi-C – for example, the approach given in main text reference [27], the approach in Supplemental Methods reference [14], the MOGEN algorithm [Trieu and Cheng, Bioinformatics 2016, <https://doi.org/10.1093/bioinformatics/btv754>], or the more physically motivated approach in [Zhang and Wolynes, PNAS 2015, <https://doi.org/10.1073/pnas.1506257112>].”*

Thank you very much for pointing out this issue. Following your advice, we have now compared our method with these suggested other methods. For ease of discussion, these methods are grouped into: Methods of Type (I), which are physically-based methods of Giorgetti *et al.* [16] (ref [27]), and the MiChroM method from Rice University (Zhang and Wolynes, 2015; Di Pierro *et al.*, 2016) [19, 26], as well as Methods of Type (II) which are two more data-driven methods of Wang *et al.* [17] (SM ref [14]) and MOGEN (Trieu and Cheng) [18]. Below we discuss these separately.

Methods (I). The two physically-based methods are excellent tools that have furthered our understanding towards the principles governing chromatin structures. The

importance of the MiChroM method was discussed in the commentary accompanying one of their key publications [26], which one of the authors of this manuscript wrote [25].

As our goals are different, our method is designed differently, and is better equipped in investigating certain issues that these other methods are not meant to. The method of [16] is an earlier study based on 5C experiments. As no results based on Hi-C data were reported, it is not possible to make direct comparison with our Hi-C based method. The MiChroM method was designed to study global scale genome folding, hence the resolution was at 50 kb. While our results on whole chromosome are comparable to that of MiChroM, as indicated by similar Pearson correlation r values (average of 0.956 for human chromosome at 50 kb for MiChroM vs. 0.94 for drosophila Chr X but at more detailed 5 kb), albeit it is more involved to model at 5 kb vs. 50 kb. A direct comparison with MiChroM at the same 5 kb or 2 kb resolution is not possible.

We have added a new section of **Comparison with other methods in 3D modeling in Supplementary Methods** to explain the differences between ours and these two methods. We first discuss each method, then point out how our method differs:

Monte Carlo model of ref [16]. In [16], an artificial square-well potential was employed, with the depth, width and location of the potential optimized through curve-fitting to the 5C-data modeled. This optimized potential was then used to generate 5,000 conformations through Metropolis Monte Carlo.

Validation was in the form of agreement with the distributions, mean, and standard deviation of pair-distances of genomic regions measured by FISH. Similar validation is provided in our study (Figure 4d), where we showed our model can reproduce the distance distributions between Scyl-chrb and distributions of two other pair-interactions. As this study is based on 5C data, no results on comparing simulation with experimental Hi-C data were reported.

MiChroM model. The MiChroM method [19, 26] investigates the energy landscape of chromatin folding. Its aim is to study general folding properties emerging from different factors in the chromatin energy model. A fully specified energy model requires A) additional information along the chromosome. Specifically, A1) beads are assigned to a specific chromatin states; and A2) the locations of loop anchors are specified (such as CTCF sites). Further, there are B) 27 parameters for the physical interactions: interaction energy of beads of different chromatin states; interaction energy of loop anchors; and distance-dependent compaction.

Both A1) and A2) are obtained based on annotation provided by Rao et al [3]. The 27 adjustable parameters are estimated from Hi-C data of chromosome 10. The model is then applied to other chromosomes, in which 2×10^4 steps of MD simulations were carried out to generate chromatin ensembles. Overall, Hi-C contact maps can be reproduced at r of 0.956. In addition, a number of important findings were made with this model, including the likely origin of chromosome territories, phase separation among chromatin types, active genes

in the periphery of chromosome territories, and knottedness of chromatin (see also a commentary on this work [25]).

As it was designed to study whole genome folding, the resolution of MiChroM is necessarily limited to 50 kb. Therefore, this model does not provide detailed spatial structures below 50 kb. At a whole chromosome level, our method has similar Pearson correlation coefficient ($r = 0.94$ for *Drosophila* chr X), but at a higher resolution of 5 kb.

Methods (II). We also examined the MOGEN method that was pointed out to us. This is a method based on optimization. The objective function is empirically constructed to enforce contacts and reduce clashes, although physical interpretations appear to be more difficult compared to MiChroM. The outcome of such an optimization procedure is connected coordinates that form a chromatin chain, with Hi-C information encoded in the objective function probabilistically satisfied through optimization.

500 whole-genome chromosome structures were obtained using Hi-C data, likely at the same resolution of 1 Mb as reported for a synthetic data set. Reported result were of coarse-grained nature, e.g., mean distances of center of mass (CoM) of individual chromosome to CoM of the whole genome. No detailed results beyond these coarse grained findings were provided, and no correlation between simulated and measured Hi-C contact maps were given. It is therefore impossible for us to directly compare our method based on Hi-C with this method.

We have added the following discussion on MOGEN to the Supplementary Methods:

MOGEN. The MOGEN method is data-driven and is based on optimization. It employs an empirical objective function designed to satisfy as many high probability restraints as possible, while reducing clashes. Restraints are converted from Hi-C data based on frequency thresholds. Parameters are adjusted so chains can be generated successfully after optimization by gradient ascent. MOGEN was used to generate a set of 500 genomewide chromosome structures using Hi-C data. However, it is unclear how well this approach works at finer base-pair resolutions (e.g. 5 kb).

In addition, we have also consulted the method of Wang et al [17] that was pointed out. This is a method based on 3C data in which each chromatin chain assumes an idealized energy (excluded volume, bending, and stretching). This energy model and 3C measurements are then combined under a Bayesian framework. The Expectation-Maximization algorithm is employed to infer unknown parameters so the likelihood function can be calculated, and the posterior probability distribution of the chromatin conformations given the 3C data can be inferred.

The energy model appears to be less well motivated than MiChroM, as no detailed physical justifications on the bending and stretching terms were provided. Validation of this method was based on FISH studies of the yeast genome, in which modeled distances are compared with FISH measurements. Similar validation is provided in

our study (Figure 4d), where we showed that our model can reproduce the distance distributions between Scyl-chrb and distributions of two other pair-interactions.

As simulated contact maps were not reported in this paper, direct comparison between this 3C-based method and our Hi-C models (at 2 kb or 5 kb resolution) is not possible. In fact, more comprehensive studies and predictions on yeast genome structures were carried out in our previous study [27], where we compared our simulated Hi-C maps with measured Hi-C map. Both inter- and intra-chromosomal Hi-C contacts can be predicted ($r = 0.82$ and $r = 0.91$, respectively). We believe our genome-wide Hi-C based models are strong results.

We have added the following to the SI:

3C-based Model. The method of Wang et al [17] was developed for analysis of 3C data, in which chromosomes are modeled as connected fragments, whose lengths are determined by restriction enzyme cuts. The idealized energy of each chromatin chain, consisting of terms for excluded volume, bending, and stretching, is combined with 3C measurements under a Bayesian framework, where parameters of a Gaussian-based likelihood function is inferred using the EM algorithm. This method is used to generate an ensemble of yeast genome structures, with the results validated by comparing distance distributions with FISH measurements.

While chromatin structures reported are at fragment resolution, there are other studies of budding yeast, where simulated and measured Hi-C heat maps show excellent correlations [17, 27, 28]. For example, our earlier ensemble models of yeast genome based on a random model similar to that of this study exhibit an excellent whole-genome correlation ($r = 0.82$) at 15 kb resolution [27].

Our contributions. To clarify our contributions, we have added the following to the main text:

Our approach for constructing 3D ensembles differs from several existing methods [16–19, 26]. With only basic physical considerations of fiber density and ligation distance threshold, it is minimalistic and there are no adjustable parameters. No chromatin states are assigned to polymer beads, and there are no *a priori* assumptions on locations of loop anchors. Furthermore, our method samples well. With deep sampling enabled through Bayesian sequential inference, our method is unique in its ability to generate a large number (e.g. 5×10^4) of diverse chromatin conformations that are consistent with Hi-C measurements.

and

With the ability to convert 2D high-resolution population Hi-C heat maps into $\sim 5.0 \times 10^4$ modeled single-cell chromatin conformations at high resolution, our method can provide quantification of chromatin heterogeneity (Fig 4c). This will allow possible differentiation of chromatin configurations that are functional from those that are not.

These points are further elaborated in the **Supplementary Methods**:

Our method. Our method differs from several other methods [16–19, 26] in important regards:

Minimal assumptions. Our model is minimalistic. It does not require assigning each bead to a specific chromosome state. There is also no *a priori* assumption on where loop anchor sites are located (e.g. CTCF). Rather, it requires only a small number of contact probabilities at different loci pairs. These are the specific interactions derived from Hi-C data analysis using our null model. With basic physical assumptions (chromatin fiber density, self-avoiding chains, and Euclidean distance threshold in nm for ligation), there are no adjustable model parameters. The target distribution from which our Hi-C ensembles are constructed is that of self-avoiding chromatin chains satisfying Hi-C derived pairing probabilities.

This minimalistic approach offers a useful advantage, namely, it avoids another level of complexity in deconvolving degeneracy due to a large parameter space, which is in addition to the complexity of deconvolving Hi-C heat maps into an ensemble of 3D chromatin configurations. In models where chromatin beads are assigned to different states and values of binding affinities are required, there are likely many permutations of states and binding affinities that are all consistent with the Hi-C data. Our model avoids this potential problem of limited identifiability.

Deep sampling. Our method also differs from several existing methods in an important aspect. We make use of a well-designed sampling distribution to improve sampling efficiency, which is dynamically adjusted based on the chromatin polymer ensemble at each growing step (Supplementary Figure S2a). The sampled ensemble rigorously follows that of the predefined target distribution, *i.e.*, self-avoiding chromatin chains satisfying Hi-C derived pairing probabilities. This in contrast to several optimization-based method (such as the method of [17]), where each chain is optimized separately through an objective function constructed on single chains (Supplementary Figure S2b).

Our sampling method is built upon significant advancements in sequential importance sampling techniques [20, 30]. As a result, we achieve very deep (*i.e.*, large-scale) sampling. For example, the ensembles we constructed contain 50,000 chromatin configurations, each of 1,000 beads of 2 kb (for loci) or 4,485 beads of 5 kb (for chromosome X), and are independently grown by sequentially adding one bead at a time. With advances in resampling and rejection control techniques far beyond the well-known PERM approach [32], we are able to overcome the well-recognized attrition problem of severe difficulty in growing long SAW chains in confined volume [22, 30]. Our sampling method is also fast. This is shown in Supplementary Figure S8d, where the averaged CPU costs (in minutes) for construction of ensembles for ten regions by specific interactions at different chain length (100–1,000 beads) are provided.

This level of deep sampling likely compares very favorably with previous studies. Under ergodic assumptions, both molecular dynamics (MD) simulations and Markov chain Monte Carlo can generate proper samples after long runs and establishment of convergence. However, achieving convergence is nontrivial, and obtaining decorrelated samples at accessible time scales along simulation trajectories is challenging in both cases. Our ensembles of 5×10^4 chromatin polymers from chain growth are results of better sampling. Compared to MD simulation, our method does not suffer from the issue of in-sufficient simulation time. Compared to MCMC simulations, our method does not suffer from possible slow convergence, which is currently an unsolved problem. In general, our method can generate more quality samples in a shorter time frame, therefore is better equipped at quantifying heterogeneity than those from 2×10^4 steps of MD simulations reported in [26], the 5,000 conformations generated using Metropolis Monte Carlo reported in [16], or typically a few hundred structures obtained using methods based on optimization [18].

Realistic ligation model. Our probabilistic model of ligation is realistic with less assumptions. We assume regions in close spatial proximity below a threshold are available to form Hi-C ligations probabilistically. Hi-C probabilities therefore represents the proportions of polymer chains that satisfy the ligation threshold (Supplementary Figure S2c). In contrast to the approach of [17], where Hi-C contact frequencies are translated to spatial distances via a reverse exponential function under implicit assumptions regarding the rate of chromatin diffusion and how it relates to the mean contact frequency, our model makes no such assumptions (Supplementary Figure S2d).

High resolution. The resolution of our model is high: 2 kb for specific loci, and 5 kb for whole chromosome, which is at the same resolution of many Hi-C studies.

Overall, our method is well-suited to a number of tasks, such as quantitative assessment of chromatin heterogeneity and determination of three-body interactions. It enabled us to reveal the novel insights reported in this study.

A new figure reporting computing time is also added to **Supplementary Figure:**

Supplementary Figure S8. (d) averaged CPU time (minutes) for constructing 3D polymer chains (50,000 chains, 20 Intel(R) Xeon(R) E5-2650 CPU cores) using frequencies of specific interactions for the 10 regions

7. *“(1) When the sequential Bayesian inference procedure is introduced in “Specific interactions are sufficient...”, the use of “constraints” to refer to the role that specific interactions play in the reconstruction was confusing to me. In my interpretation, the role they play is to require certain contact frequencies averaged over the ensemble, but the use of the word constraints suggests that each member of the reconstruction is being constrained to have those loci in contact. My preference is that constraints be replaced by phrasing involving the word “frequency”.*”

We apologized for the confusion. We have changed the words “constraints” to “frequencies”.

“(2) In general, a high-level explanation of the method would be appreciated here, as it doesn’t appear that any is currently given; readers must refer to the Supplementary Methods in order to understand basics of the method. Given that several methods exist in the literature which purport to reconstruct ensembles of chromatin structures based on Hi-C, it would also help the reader in terms of understanding the precise way in which the proposed method is novel.”

Thank you for this helpful suggestion. To explain our method, we have now added a new section providing **Overview of our approach to 3D chromatin modeling in Results:**

Our method relies on the recently developed capability of deep sampling to generate 3D chromatin ensembles [20,22,23]. We first identify a set of *specific chromatin interactions* from deep-sequenced population Hi-C data. They are unlikely due to ligation of randomly collided loci and are identified by comparing measured Hi-C frequencies to those simulated from an ensemble of randomly folded 3D chromatin configurations. The specific interactions identified are then used as restraints to generate an ensemble of modeled single-cell chromatin configurations for either a genomic region (200 kb to 4 Mb at 2 kb resolution) or a whole chromosome (22 Mb, at 5 kb resolution). Structural analysis, including Euclidean distance measurement and spatial clustering, is then carried out on the ensemble of modeled single-cell chromatin configurations. This provides an overall quantitative assessment of population 3D structural properties, such as compactness, radius of gyration, domain boundaries, as well as chromatin structural heterogeneity. Results from structural analysis are then integrated with other information such as enhancer/promoter information and epigenetic modifications for additional biological insight.

We have also modified Fig 1 to provide more details of our method, and have added the following to the section **Specific interactions constitute a small fraction of Hi-C contacts** in the main text:

Briefly, for our null model, we generate multiple chromatin chains through a fractal Monte Carlo method based on chain-growth [20, 22, 23, 31], in which monomers are added one at a time. To overcome severe difficulties in sampling,

this process is optimized through a recursive resampling algorithm applied at predetermined check-point lengths [20, 22, 23]. The target distribution of the ensemble is overall geometrically realizable self-avoiding chromatin chains within the specified volume confinement. This target distribution is rigorously enforced through proper importance weighting during chain-growth (see Supplementary Methods for more details).

and to the section **Specific interactions are sufficient to drive chromatin folding** in the main text:

For ensembles constrained by Hi-C interactions, we derive contact probabilities from Hi-C frequencies. The target distribution of the chromatin ensemble is that of self-avoiding chromatin chains that satisfy the Hi-C derived contact probabilities at each chain-growing step, where beads are placed sequentially for each chain. Our sequential Bayesian inference framework ensures that the chromatin chains are consistent with the observed Hi-C, and ensemble properties can be accurately estimated (Supplementary Figure S7).

We have also added a new figure as Supplementary Figure S2 to illustrate our method:

Supplementary Figure S2. An illustration of our method based on chain growth and comparison with other methods. **(a)** Chromatin polymer ensemble is constructed sequentially, with the addition of one bead at a time. We make use of an optimized sampling distribution at each step to improve sampling efficiency, which is dynamically adjusted after adding each bead. **(b)** A data-driven method for construction of chromatin polymer ensemble as found in [18] and [17]. All beads are placed at once during the initialization step. Each simulation generates a chromatin chain through a separate optimization procedure, where the object function is constructed individually for each single chain. **(c)** In our model, Hi-C probability corresponds to the proportion of polymer chains that satisfy the ligation threshold. **(d)** In several other methods, Hi-C probability is transformed into a spatial distance via a reverse function [17].

8. *“The authors follow changes in contacts between genomic regions identified as belonging to A (active), I (inactive), and P (polycomb). The assignment of these regions to a particular domain type is based on ChIP-seq data. Perhaps this is known, but evidence that these assignments do not change over the course of development is necessary, in my view. Otherwise, these changes in contact frequencies could be attributed to changes in domain assignment.”*

Thanks for pointing out this important issue. To confirm chromatin states identified from epigenetic modifications of S2 cells are stable across different developmental stages, we have followed your advice and carried out additional studies.

We have added the new results on this to **Supplementary Methods**:

To assess possible changes in the distribution of different interaction types, we use the 15 markers of S2 to categorize all specific interactions from each cell type, as the chromatin classification based on S2 cells is robust across embryos at cycle 12 and cycle 14c. This is evidenced by an analysis of three representative histone modifications (H3K37me3, H3K36me3 and H3K4me3) following the approach of [4]. Supplementary Figure S3 shows that after signals of histone modifications in 2 kb bins are aggregated over the same three chromatin states (Active, Inactive and Polycomb-repressed) defined by S2 markers, their distributions are largely unchanged across embryos at cycle 12 (during cycles 9–13) and cycle 14c (during stages 5–8).

We have also added a corresponding Supplementary Figure S5:

Supplementary Figure S5. Distributions of H3K27me3, H3K36me3 and H3K4me3 ChIP-seq or ChIP-chip quantile-normalized signals in 2 kb bins from embryos at cycle 12, cycle 14c, and S2-DRSC. The three chromatin states (Active, Inactive and Polycomb-repressed) are assigned based on S2 markers. ChIP-seq and ChIP-chip data are taken from the GEO database (Data sources listed in Supplementary Table S3). Processed wiggle files downloaded from the database are quantile-normalized before comparison.

9. “In the section demonstrating three body interactions, it would again be good to know that the sequential Bayesian framework is capable of observing changes in three-body interaction frequencies in silico, by construction of an ensemble with a particular frequency of three body interactions which is then inferred by the proposed method. The FISH comparison in Fig. S4g is a nice step, but I think it’s important that the extent of quantitative success of the method is explored in a more controlled environment.”

Thank you for your very helpful advice. We can now address this issue with results obtained when answering Question 3 raised earlier. Specifically, we have examined the spatial distance distributions among three anchors $p1$, $p2$, and $p3$ in a simulated region containing a unit of three-body interactions. The distribution of distances $p1-p2$, $p1-p3$ and $p2-p3$ are overall very similar between the original and the reconstructed ensembles (see Supplementary Figure S7h shown below). Moreover, the fractions of the loop pairs and the three-body contacts $p1-p2-p3$ are highly consistent between those from the original and those from the reconstructed ensembles (Supplementary Figure S7i).

We have added the following sentence to **Supplementary Methods**:

Our approach can uncover 3-body interaction unit, as shown in Supplementary Figure S7h and S7i.

Supplementary Figure S7. (h) Distance distributions of $p1-p2$, $p1-p3$, and $p2-p3$ looping contact pairs labeled in (a) for the original and reconstructed ensembles. (i) Fractions of $p1-p2$, $p1-p3$, $p2-p3$ looping pairs and $p1-p2-p3$ three-body contact pairs in the original and reconstructed ensembles.

10. “Given that both the original and simulated Hi-C maps in Fig 5a seem essentially featureless, the authors’ claim to have identified TAD-like structures is surprising and interesting. This leads to two separate questions:

(1) Is there perhaps a signal of TAD structures in the bulk Hi-C map? The authors can quantitatively assess this using an insulation or boundary strength score; it seems like the current evaluation is mostly qualitative. ”

Thank you for this suggestion. We have now calculated the *Directionality Index* (DI) for the region of chr2L: 11.0–12.0 Mb to examine how boundary strength varies among the three cell types following ref [15]. We have added the following text and an accompanying figure (Supplementary Figure S11b) to the new section **Evaluation of boundary strength by Directionality Index** in **Supplementary Methods**:

We also evaluated the boundary strength directly using population Hi-C contact matrices. We calculated the Directionality Index (DI) for each 2 kb bin following ref [15]. DI is calculated as:

$$DI = \left(\frac{f_{down} - f_{up}}{|f_{down} - f_{up}|} \right) \left(\frac{(f_{up} - f_{exp})^2}{f_{exp}} + \frac{(f_{down} - f_{exp})^2}{f_{exp}} \right),$$

where f_{up} is the contact frequency between the given 2 kb bin and the upstream 100 kb region; f_{down} is the contact frequency between the same bin and the downstream 100 kb region; f_{exp} is the expected contact frequency, which is set to $\frac{f_{up} + f_{down}}{2}$. We find that embryos at post-MBT (stages 5–8) and S2R+ show strong DI signals, indicating clear TAD structures with sharp boundaries. In contrast, DI signals are weak or non-existent in early embryos at pre-MBT stages (cycles 9–13) (Supplementary Figure S11b), indicating lack of TAD structures as judged from population Hi-C contact matrix (Figure 5a).

“(2) To what extent is the ensemble constructed simply a reflection of random fluctuations? Said another way, if the authors were to construct an ensemble of SARWs, would they also observe a similar subpopulation of configurations with TAD-like structures? Said yet another way, can the authors provide a convincing null model here?”

Thank you for raising this very helpful question. We have further investigated the domain structures in chromatin configurations in a random ensemble. Our results are now reported in the section of **Identification of single-cell domain boundaries and TAD-like structures** of the **Supplementary Methods**:

Similar to earlier findings [22], many chromatin configurations in a control random ensemble of chr2L: 11.0–12.0 Mb also exhibit TAD-like structures: 23.1% are found to maintain at least one TAD-like structure. However, this is significantly less compared to 54.4% of chromatin configurations in embryos at pre-MBT stages (cycles 9–13) (Figure 5a). Furthermore, the TAD boundaries are distributed very evenly in the random ensemble, in contrast to cells at pre-MBT and later stages (Supplementary Figure S11a). These results suggest that

while TAD-like structures may form in random chromatin configurations, other biological factors beyond fluctuation play significant roles during TAD formation in early embryos.

Supplementary Figure S11. Simulated single-cell domain boundaries among cells at three developmental stages of embryogenesis (pre-MBT (cycles 9–13), post-MBT (stages 5–8) and S2R+ derived from late embryos). **(b)** Directionality Index (DI) for cells at the three different stages in the region of chr2L: 11.0 – 12.0 Mb shown in Figure 5a.

Supplementary Figure S11. (a) Distributions of domain boundary probabilities along genomic positions of the region shown in Figure 5A (chr2L: 11.0–12.0 Mb) in the three cell types and the random polymer ensemble. All ensembles consists of 5,000 configurations. The diminished random probabilities at the start and end of the region are boundary effects and can be corrected.

References

- [1] Ferhat Ay, Timothy L Bailey, and William Stafford Noble. Statistical confidence estimation for hi-c data reveals regulatory chromatin contacts. *Genome research*, 24(6):999–1011, 2014.
- [2] Borbala Mifsud, Inigo Martincorena, Elodie Darbo, Robert Sugar, Stefan Schoenfelder, Peter Fraser, and Nicholas M Luscombe. Gothic, a probabilistic model to resolve complex biases and to identify real interactions in hi-c data. *PloS one*, 12(4):e0174744, 2017.
- [3] Suhas SP Rao, Miriam H Huntley, Neva C Durand, Elena K Stamenova, Ivan D Bochkov, James T Robinson, Adrian L Sanborn, Ido Machol, Arina D Omer, Eric S Lander, et al. A 3d map of the human genome at kilobase resolution reveals principles of chromatin looping. *Cell*, 159(7):1665–1680, 2014.
- [4] Yuki Ogiyama, Bernd Schuettengruber, Giorgio L Papadopoulos, Jia-Ming Chang, and Giacomo Cavalli. Polycomb-dependent chromatin looping contributes to gene silencing during drosophila development. *Molecular cell*, 71(1):73–88, 2018.
- [5] Clemens B Hug, Alexis G Grimaldi, Kai Kruse, and Juan M Vaquerizas. Chromatin architecture emerges during zygotic genome activation independent of transcription. *Cell*, 169(2):216–228, 2017.
- [6] Bogdan Bintu, Leslie J Mateo, Jun-Han Su, Nicholas A Sinnott-Armstrong, Mirae Parker, Seon Kinrot, Kei Yamaya, Alistair N Boettiger, and Xiaowei Zhuang. Super-resolution chromatin tracing reveals domains and cooperative interactions in single cells. *Science*, 362(6413):eaau1783, 2018.
- [7] Marshall Fixman. Radius of gyration of polymer chains. *The Journal of Chemical Physics*, 36(2):306–310, 1962.
- [8] Yad Ghavi-Helm, Felix A Klein, Tibor Pakozdi, Lucia Ciglar, Daan Noordermeer, Wolfgang Huber, and Eileen EM Furlong. Enhancer loops appear stable during development and are associated with paused polymerase. *Nature*, 512(7512):96, 2014.
- [9] Jeffrey A Farrell and Patrick H O’Farrell. From egg to gastrula: how the cell cycle is remodeled during the drosophila mid-blastula transition. *Annual review of genetics*, 48:269–294, 2014.
- [10] Chunhui Hou, Li Li, Zhaohui S Qin, and Victor G Corces. Gene density, transcription, and insulators contribute to the partition of the drosophila genome into physical domains. *Molecular cell*, 48(3):471–484, 2012.
- [11] Sergey V Uljanov, Ekaterina E Khrameeva, Alexey A Gavrillov, Ilya M Flyamer, Pavel Kos, Elena A Mikhaleva, Aleksey A Penin, Maria D Logacheva, Maxim V Imakaev, Alexander Chertovich, et al. Active chromatin and transcription play a key role in chromosome partitioning into topologically associating domains. *Genome research*, 26(1):70–84, 2016.

- [12] Qi Wang, Qiu Sun, Daniel M Czajkowsky, and Zhifeng Shao. Sub-kb hi-c in *d. melanogaster* reveals conserved characteristics of tads between insect and mammalian cells. *Nature communications*, 9(1):188, 2018.
- [13] Jutta Vogelmann, Antoine Le Gall, Stephanie Dejardin, Frederic Allemand, Adrien Gamot, Gilles Labesse, Olivier Cuvier, Nicolas Negre, Martin Cohen-Gonsaud, Emmanuel Margeat, et al. Chromatin insulator factors involved in long-range dna interactions and their role in the folding of the drosophila genome. *PLoS genetics*, 10(8):e1004544, 2014.
- [14] Simona Bianco, Darío G Lupiáñez, Andrea M Chiariello, Carlo Annunziatella, Katerina Kraft, Robert Schöpflin, Lars Wittler, Guillaume Andrey, Martin Vingron, Ana Pombo, et al. Polymer physics predicts the effects of structural variants on chromatin architecture. *Nature genetics*, 50(5):662, 2018.
- [15] Jesse R Dixon, Siddarth Selvaraj, Feng Yue, Audrey Kim, Yan Li, Yin Shen, Ming Hu, Jun S Liu, and Bing Ren. Topological domains in mammalian genomes identified by analysis of chromatin interactions. *Nature*, 485(7398):376, 2012.
- [16] Luca Giorgetti, Rafael Galupa, Elphège P Nora, Tristan Piolot, France Lam, Job Dekker, Guido Tiana, and Edith Heard. Predictive polymer modeling reveals coupled fluctuations in chromosome conformation and transcription. *Cell*, 157(4):950–963, 2014.
- [17] Siyu Wang, Jinbo Xu, and Jianyang Zeng. Inferential modeling of 3d chromatin structure. *Nucleic acids research*, 43(8):e54–e54, 2015.
- [18] Tuan Trieu and Jianlin Cheng. Mogen: a tool for reconstructing 3d models of genomes from chromosomal conformation capturing data. *Bioinformatics*, 32(9):1286–1292, 2015.
- [19] Bin Zhang and Peter G Wolynes. Topology, structures, and energy landscapes of human chromosomes. *Proceedings of the National Academy of Sciences*, 112(19):6062–6067, 2015.
- [20] Alan Perez-Rathke, Qiu Sun, Boshen Wang, Valentina Boeva, Zhifeng Shao, and Jie Liang. Chromatix: computing the functional landscape of many-body chromatin interactions in transcriptionally active loci from deconvolved single cells. *Genome Biology*, 21(1):1–17, 2020.
- [21] Andrew S Belmont. Large-scale chromatin organization: the good, the surprising, and the still perplexing. *Current opinion in cell biology*, 26:69–78, 2014.
- [22] Gamze Gürsoy, Yun Xu, Amy L Kenter, and Jie Liang. Spatial confinement is a major determinant of the folding landscape of human chromosomes. *Nucleic acids research*, 42(13):8223–8230, 2014.

- [23] Gamze Gürsoy, Yun Xu, Amy L Kenter, and Jie Liang. Computational construction of 3d chromatin ensembles and prediction of functional interactions of alpha-globin locus from 5c data. *Nucleic acids research*, 45(20):11547–11558, 2017.
- [24] Adrian L Sanborn, Suhas SP Rao, Su-Chen Huang, Neva C Durand, Miriam H Huntley, Andrew I Jewett, Ivan D Bochkov, Dharmaraj Chinnappan, Ashok Cutkosky, Jian Li, et al. Chromatin extrusion explains key features of loop and domain formation in wild-type and engineered genomes. *Proceedings of the National Academy of Sciences*, 112(47):E6456–E6465, 2015.
- [25] Gamze Gürsoy and Jie Liang. Three-dimensional chromosome structures from energy landscape. *Proceedings of the National Academy of Sciences*, 113(43):11991–11993, 2016.
- [26] Michele Di Pierro, Bin Zhang, Erez Lieberman Aiden, Peter G Wolynes, and José N Onuchic. Transferable model for chromosome architecture. *Proceedings of the National Academy of Sciences*, 113(43):12168–12173, 2016.
- [27] Gamze Gürsoy, Yun Xu, and Jie Liang. Spatial organization of the budding yeast genome in the cell nucleus and identification of specific chromatin interactions from multi-chromosome constrained chromatin model. *PLoS computational biology*, 13(7):e1005658, 2017.
- [28] Harianto Tjong, Ke Gong, Lin Chen, and Frank Alber. Physical tethering and volume exclusion determine higher-order genome organization in budding yeast. *Genome research*, 22(7):1295–1305, 2012.
- [29] Hua Wong, Hervé Marie-Nelly, Sébastien Herbert, Pascal Carrivain, Hervé Blanc, Romain Koszul, Emmanuelle Fabre, and Christophe Zimmer. A predictive computational model of the dynamic 3d interphase yeast nucleus. *Current biology*, 22(20):1881–1890, 2012.
- [30] Jun S Liu. *Monte Carlo strategies in scientific computing*. Springer Science & Business Media, 2008.
- [31] Jinfeng Zhang, Rong Chen, Chao Tang, and Jie Liang. Origin of scaling behavior of protein packing density: A sequential monte carlo study of compact long chain polymers. *The Journal of chemical physics*, 118(13):6102–6109, 2003.
- [32] Hsiao-Ping Hsu and Peter Grassberger. A review of monte carlo simulations of polymers with perm. *Journal of statistical physics*, 144(3):597, 2011.
- [33] Ivan Junier, Yannick G Spill, Marc A Marti-Renom, Miguel Beato, and François le Dily. On the demultiplexing of chromosome capture conformation data. *FEBS letters*, 589(20):3005–3013, 2015.
- [34] Anne Scuderi, Karl Simin, Sandra G Kazuko, James E Metherall, and Anthea Letsou. scylla and charybde, homologues of the human apoptotic gene rtp801, are required for head involution in drosophila. *Developmental biology*, 291(1):110–122, 2006.

- [35] Tom Sexton, Eitan Yaffe, Ephraim Kenigsberg, Frédéric Bantignies, Benjamin Leblanc, Michael Hoichman, Hugues Parrinello, Amos Tanay, and Giacomo Cavalli. Three-dimensional folding and functional organization principles of the drosophila genome. *Cell*, 148(3):458–472, 2012.
- [36] Fulai Jin, Yan Li, Jesse R Dixon, Siddarth Selvaraj, Zhen Ye, Ah Young Lee, Chia-An Yen, Anthony D Schmitt, Celso A Espinoza, and Bing Ren. A high-resolution map of the three-dimensional chromatin interactome in human cells. *Nature*, 503(7475):290, 2013.
- [37] Takashi Nagano, Yaniv Lubling, Tim J Stevens, Stefan Schoenfelder, Eitan Yaffe, Wendy Dean, Ernest D Laue, Amos Tanay, and Peter Fraser. Single-cell hi-c reveals cell-to-cell variability in chromosome structure. *Nature*, 502(7469):59, 2013.
- [38] Takashi Nagano, Yaniv Lubling, Csilla Várnai, Carmel Dudley, Wing Leung, Yael Baran, Netta Mendelson Cohen, Steven Wingett, Peter Fraser, and Amos Tanay. Cell-cycle dynamics of chromosomal organization at single-cell resolution. *Nature*, 547(7661):61, 2017.
- [39] Ilya M Flyamer, Johanna Gassler, Maxim Imakaev, Hugo B Brandão, Sergey V Ulianov, Nezar Abdennur, Sergey V Razin, Leonid A Mirny, and Kikuë Tachibana-Konwalski. Single-nucleus hi-c reveals unique chromatin reorganization at oocyte-to-zygote transition. *Nature*, 544(7648):110, 2017.
- [40] Tim J Stevens, David Lando, Srinjan Basu, Liam P Atkinson, Yang Cao, Steven F Lee, Martin Leeb, Kai J Wohlfahrt, Wayne Boucher, Aoife O’Shaughnessy-Kirwan, et al. 3d structures of individual mammalian genomes studied by single-cell hi-c. *Nature*, 544(7648):59, 2017.
- [41] Siyuan Wang, Jun-Han Su, Brian J Beliveau, Bogdan Bintu, Jeffrey R Moffitt, Chaoting Wu, and Xiaowei Zhuang. Spatial organization of chromatin domains and compartments in single chromosomes. *Science*, 353(6299):598–602, 2016.
- [42] Quentin Szabo, Daniel Jost, Jia-Ming Chang, Diego I Cattoni, Giorgio L Papadopoulos, Boyan Bonev, Tom Sexton, Julian Gurgo, Caroline Jacquier, Marcelo Nollmann, et al. Tads are 3d structural units of higher-order chromosome organization in drosophila. *Science advances*, 4(2):eaar8082, 2018.
- [43] Leslie J Mateo, Sedona E Murphy, Antonina Hafner, Isaac S Cinquini, Carly A Walker, and Alistair N Boettiger. Visualizing dna folding and rna in embryos at single-cell resolution. *Nature*, 568(7750):49, 2019.
- [44] Jie Liang, Jinfeng Zhang, and Rong Chen. Statistical geometry of packing defects of lattice chain polymer from enumeration and sequential monte carlo method. *Journal of Chemical Physics*, 117(7):3511–3521, 2002.
- [45] Yoav Benjamini, Daniel Yekutieli, et al. The control of the false discovery rate in multiple testing under dependency. *The annals of statistics*, 29(4):1165–1188, 2001.

REVIEWERS' COMMENTS:

Reviewer #1 (Remarks to the Author):

The revised manuscript by Sun et al. is much clearer, includes a much better explanation of the method, sufficiently acknowledges previous work, and is overall significantly improved from the initial submission. The authors have satisfactorily responded to all of my previous suggestions, and overall appear to have put in a significant amount of effort to respond to the comments made by all reviewers. I have only a couple of additional, non-essential, suggestions that I believe could potentially improve the manuscript:

1. Hug et al. (PMID: 28388407) identified some boundaries in pre-MBT embryos that are associated with genes that are expressed zygotically before MBT (including at the Bsg25A locus). Are there pre-MBT boundary location preferences in modelled conformations of these loci?
2. There are a couple of studies on *Drosophila* embryos that have compared high resolution imaging data with Hi-C from comparable embryo stages (e.g. Mateo et al. (PMID: 30886393) and Cardozo Gizzi et al. (PMID: 30795893)). These could provide an excellent opportunity to examine whether the populations of modelled single cell conformations from this method are consistent with single-cell imaging data, without requiring new experiments. While the revised manuscript is careful to avoid making the claim that the modelled single cell conformations reflect real conformations found in embryos, this comparison would help to support the biological relevance of the modelled conformations.
3. Minor point: I found it difficult to identify which studies the Hi-C data is taken from by reading the main text. I would suggest adding references to these studies in the sentence describing the data in lines 127-129.

Reviewer #2 (Remarks to the Author):

The authors have satisfactorily addressed my prior comments on the methodology used and its validation.

Reviewer #3 (Remarks to the Author):

Overall, I think the paper is greatly improved, and I thank the authors for their work in accommodating the reviewer's responses.

First among the outstanding issues is the organization of the results section, which I found difficult to follow. I think the paper could be improved if the authors clarified the goal of the results sections up until line 323. It currently feels like many different ideas are being discussed simultaneously in this portion of the text. For example, Fig 1 is split across multiple sections within the text, with Fig 2 intervening. Sections corroborating the proposed method by its ability to recapitulate known biological trends, sections discussing the role specific interactions play in the strength of the proposed method, and sections explaining the method itself would ideally be clearly demarcated.

Minor comment 1: line 177 "obscure" should be "obscured" I believe.

Minor comment 2: Line 321 "Such quantitative measures of change in compactness can only be derived from physical models of 3D chromatin conformations." seems to be a bit of an overstatement. Presumably this is meant mostly within the context of the 3C-based methods?

In addition to a need for structural clarification in the first two thirds of the paper, I have some questions in regards to issues 9 and 10 from our previous correspondence:

9. Perhaps I'm misunderstanding, but the ground-truth model seems to only involve specifying

pairwise interactions. If this is the case, it is my understanding that it would not be surprising for the method to be able to capture the resulting three-body interaction frequency. Would it be possible, for instance, for the method to recover a scenario where the three points only interacted as a triple?

I also can't seem to find the exact numbers for the p1-p2, and p2-p3 frequencies discussed in the supporting text. And what is the meaning of "fraction" in Fig S7i? Is it meant to be a percent?

10. I am not sure I understand the message of the section "A large portion of single cells likely contain TAD-like structures in early embryos." The modeled boundary probabilities in pre-MBT cells appear to have correlated peaks with peaks in post-MBT and S2R+ cells, visibly over any sort of fluctuations in the random ensemble (Fig S11a). Additionally, while the magnitude of the directionality index is much smaller for pre-MBT cells, it appears to display a good correlation in sign with the DI in post-MBT and S2R+ (Fig S11b). To me, this suggests that the modelling procedure is able to pick up on an existing but weak TAD presence, as opposed to revealing the presence of single-cell TADs in a population of cells where there are no domain boundaries on aggregate. In this context, line 391 - "These results suggest that single-cell TAD-like structures could exist during *Drosophila* embryogenesis, even at the early developmental stage, where no clear TAD structures could be seen in population Hi-C studies [33]." - seems overstated.

Reviewer 1

“(General comment) The revised manuscript by Sun et al. is much clearer, includes a much better explanation of the method, sufficiently acknowledges previous work, and is overall significantly improved from the initial submission. The authors have satisfactorily responded to all of my previous suggestions, and overall appear to have put in a significant amount of effort to respond to the comments made by all reviewers.

Thank you for your kind words!

1. *“Hug et al. (PMID: 28388407) identified some boundaries in pre-MBT embryos that are associated with genes that are expressed zygotically before MBT (including at the Bsg25A locus). Are there pre-MBT boundary location preferences in modelled conformations of these loci? ”*

We have carried out additional modeling studies following your suggestion. To compare with Hug et al, we have estimated boundary probabilities along the locus of chr2L:4.5-6.5 Mb using our ensemble of single-cell chromatin conformations. Overall, our modeled boundary probabilities are in excellent agreement with both the reported insulation scores and the RNA Pol II chip signals of Hug et al (see Supplementary Figure 11d).

We have added the following text to our manuscript:

Moreover, our results are consistent with a recent study by Hug et al [2], where reported boundaries in pre-MBT embryos are associated with genes expressed zygotically before MBT. The boundary probabilities calculated from our predicted ensemble of single-cell conformations for the locus chr2L:4.5-6.4 Mb are in excellent agreement with data reported in [2] (Supplementary Figure 11d).

and the following to the Supplementary Methods:

Comparison with Hug et al.

We compared our modeled ensemble of single-cell conformations for the region of chr2L:4.5-6.4 Mb with the study of Hug et al [2]. Supplementary Figure 11d shows the boundary probabilities estimated from our ensemble (see third track), which are constructed from the Hi-C data of embryos at cycles 9–13 [4]. Overall, our estimated boundary probabilities agree well with the insulation scores and RNA Pol II ChIP-Seq signal derived from Hug et al [2].

2. *“There are a couple of studies on Drosophila embryos that have compared high resolution imaging data with Hi-C from comparable embryo stages (e.g. Mateo et al. (PMID: 30886393) and Cardozo Gizzi et al. (PMID: 30795893)). These could provide an excellent opportunity to examine whether the populations of modelled single cell conformations from this method are consistent with single-cell imaging data, without requiring new experiments. While the revised manuscript is careful to avoid making the claim that the modelled single cell conformations reflect real conformations found in*

Supplementary Figure 11 (d) Preference of simulated single-cell domain boundaries at pre-MBT boundary positions in Hi-C data of Hug et al. The heat map, insulation and RNA pol II tracks are from Hug et al (with permission). The bottom track shows the probabilities of simulated single-cell domain boundaries in the same region of chr2L: 4.5 – 6.5 Mb.

embryos, this comparison would help to support the biological relevance of the modelled conformations.”

We have now carried out additional simulation studies to compare our model with that obtained by the imaging technique of Mateo et al. The new results have been added to our manuscript - thank you for this very helpful suggestion! The new sentence added reads:

Our modeled single-cell conformations also agree well with single-cell chromatin reconstructed from another imaging study [3] (see Figure 5g and Supplementary Methods).

Figure 5. (g) Correspondence between distance maps of two modeled single-cell conformations (chr3R:12.20—12.90 Mb of *Drosophila* S2R+ at 2 kb resolution) with two conformations constructed from imaging studies in Mateo et al ($r = 0.75$ and 0.79 , Fig 2d of [3] reprinted with permission).

and a new paragraph in the Supplementary Methods:

Comparison of modeled and measured single-cell conformations.

We generated 10,000 single-cell conformations of the locus chr3R:12.20—12.90 Mb using Hi-C data of S2R+ cells. The resulting distance maps are compared pixel-wise with that depicted in Mateo et al [3], where 19,103 cells were measured, and two depicted. Despite the gap regions in reported results in Mateo et al, which makes direct comparison difficult, two single-cell conformations we modeled have overall excellent agreement with the two conformations depicted in Mateo et al, with $r = 0.75$ and 0.79 , respectively.

Regarding the Gizzi et al study, as the depicted distance maps in pixels are similar to that of Mateo et al but less challenging (< 400 kb vs. 700 kb), we decided not to pursue a detailed direct comparison due to the limited time for revision. Although we are unable to access the original raw distance data in a form that can be used for a more direct comparison, we believe our reported pixel-wise comparison with Mateo et al is adequate in illustrating the overall agreement between single-cell modeling and measurement.

3. *“Minor point: I found it difficult to identify which studies the Hi-C data is taken from by reading the main text. I would suggest adding references to these studies in the sentence describing the data in lines 127-129 ”*

We have now added these references.

Reviewer 3

1. *“First among the outstanding issues is the organization of the results section, which I found difficult to follow. I think the paper could be improved if the authors clarified the goal of the results sections up until line 323. It currently feels like many different ideas are being discussed simultaneously in this portion of the text. For example, Fig 1 is split across multiple sections within the text, with Fig 2 intervening. Sections corroborating the proposed method by its ability to recapitulate known biological trends, sections discussing the role specific interactions play in the strength of the proposed method, and sections explaining the method itself would ideally be clearly demarcated.”*

We have now reorganized this part of the text, so the clarity of the **Results** sections has been improved. The focus of Fig 1 is now the depiction of the null model and how specific interactions are identified. This figure is no longer split across different sections. Fig 3 now contains the illustration on how chromatin folding is modeled under predicted specific interactions, as well as depictions of different outcomes of folding experiments using predicted specific interactions and other types of interactions.

We have also renamed the relevant subsections up to line 323 in **Results** to provide overall demarcation of the content. These are now all under 60 space-characters as required. They now read: **“Modeling uncovers a small set of specific interactions”**, **“Specific interactions recapture known long-range interactions”**, **“Specific interactions reveal biologically-relevant patterns”**, **“Specific interactions can drive chromatin folding”**, and **“Single-cell conformations quantify chromatin heterogeneity”**.

2. *“line 177 “obscure” should be “obscured” I believe”*

This is now fixed. Thank you!

3. *““Such quantitative measures of change in compactness can only be derived from physical models of 3D chromatin conformations.” seems to be a bit of an overstatement. Presumably this is meant mostly within the context of the 3C-based methods?”*

Indeed, this statement is misleading. We have modified our statement so it now reads:

Such quantitative measures of change in compactness can only be derived from Hi-C and 3C-derived data via physical modeling of 3D chromatin conformations.

4. *“Perhaps I’m misunderstanding, but the ground-truth model seems to only involve specifying pairwise interactions. If this is the case, it is my understanding that it would not be surprising for the method to be able to capture the resulting three-body interaction frequency. Would it be possible, for instance, for the method to recover a scenario where the three points only interacted as a triple?”*

We have carried out further simulations using a toy model, where interactions among three bodies are given *a priori*. That is, the interactions among loci p1, p2, and

p3 are known to exist solely as a triplet interaction. This triplet interaction p1-p2-p3 is equivalent to enforcing the pairwise interactions p1-p2, p2-p3, and p1-p3 to simultaneously co-exist within corresponding single cells of the ensemble. Our method can indeed produce ensembles of single-cell conformations following this prescribed condition.

We have added the following paragraph to the Supporting Information:

Our model can also incorporate *a priori* information where three bodies p1, p2, and p3 are known to interact solely as a triplet. Our method can readily generate ensembles that conform to such specifications by knocking-in these simultaneous interactions [1]. This is shown in Supplementary Figure 7j with a toy example of a 4-cell ensemble. Here the simultaneous all-to-all 3-body interaction (p1-p2-p3) is given *a priori* and modeled through explicit knock-in perturbations using our CHROMATIX method [1]. Supplementary Figure 7j shows that the simultaneous 3-body interaction (colored as green, yellow, and purple) is recovered in all four cells. Such *a priori* information may be found, for example, from 3D-FISH studies and can be incorporated to complement Hi-C data using our full suite of computational tools. For full details, please see “Chromatin-to-chromatin proximity interaction modeling tutorial” available at <https://bitbucket.org/aperezrathke/chr-folder/>.

and a new figure:

Supplementary Figure 7. (j) A toy ensemble of four chromatin polymer chains generated under the *a priori* condition that the three loci p1, p2, and p3 interact solely as a triplet p1-p2-p3. The resulting ensemble is generated using knock-in perturbations of the CHROMATIX method [1], with loci p1, p2, and p3 shown as green, yellow, and purple beads, respectively.

5. *“I also can’t seem to find the exact numbers for the p1-p2, and p2-p3 frequencies discussed in the supporting text. And what is the meaning of “fraction” in Fig S7i? Is it meant to be a percent? ”*

We have now added the ensemble size information (*i.e.*, the number of chromatin chains) to the caption of Fig S7 for the original and the reconstructed polymer ensembles, and replaced the word “fraction” with “percentage”:

(a) First, we construct an original chromatin polymer ensemble **consisting of 50,000 chains** and aggregate the single-chain contact maps into a simulated Hi-C contact map; we then apply our sequential Bayesian inference framework and reconstruct another polymer ensemble **consisting of 50,000 chains** solely from the simulated Hi-C contact map.

(i) **Percentages** of p1-p2, p1-p3, p2-p3 looping pairs and p1-p2-p3 three-body contact pairs in the original and reconstructed ensembles.

The values of % of p1-p2 and p2-p3 have also been added to the bar-plot of Supplementary Figure 7i.

The following sentence has also been added to the relevant section of the **Supplementary Methods**:

Further quantitative information can be found in the caption of Supplementary Figure 7.

6. *“I am not sure I understand the message of the section “A large portion of single cells likely contain TAD-like structures in early embryos.” The modeled boundary probabilities in pre-MBT cells appear to have correlated peaks with peaks in post-MBT and S2R+ cells, visibly over any sort of fluctuations in the random ensemble (Fig S11a). Additionally, while the magnitude of the directionality index is much smaller for pre-MBT cells, it appears to display a good correlation in sign with the DI in post-MBT and S2R+ (Fig S11b). To me, this suggests that the modelling procedure is able to pick up on an existing but weak TAD presence, as opposed to revealing the presence of single-cell TADs in a population of cells where there are no domain boundaries on aggregate. In this context, line 391 – “These results suggest that single-cell TAD-like structures could exist during Drosophila embryogenesis, even at the early developmental stage, where no clear TAD structures could be seen in population Hi-C studies [33].” – seems overstated. ”*

Thank you for pointing out this issue. Indeed, there are weak signals of TAD presence in the Hi-C data. In fact, such signals are the basis upon which our method successfully built 3D models of single-cell chromatin conformations, a subset of which exhibit TAD-like structures (e.g. Figure 5c). However, the signal for such a subpopulation with TAD-like structure gets averaged out when the direction index is calculated using population Hi-C data. It is therefore difficult to draw definitive conclusions by examining the population Hi-C data alone, where tiny traces of peaks are present in the direction index plots (Supplementary Figure 11b, stack 1). The strength of the

signal in pre-MBT is nowhere near the strong peaks we see in post-MBT and S2R+ cells. Overall, it would be challenging to definitively declare the presence of TADs among a subpopulation of cells using a population-averaged direction index. This is in contrast to post-MBT and S2R+ cells, where the direction index signal is strong and unambiguous (Supplementary Figure 11b, stacks 2 and 3).

We have modified line 391 to avoid possible misleading interpretation:

These results suggest that single-cell TAD-like structures could exist during *Drosophila* embryogenesis, even at the early developmental stage, where TAD structures could not be clearly detected with confidence from population-averaged Hi-C data (see Supplementary Figure S11b row 1, in contrast to rows 2 and 3 [4]).

References

- [1] Alan Perez-Rathke, Qiu Sun, Boshen Wang, Valentina Boeva, Zhifeng Shao, and Jie Liang. Chromatix: computing the functional landscape of many-body chromatin interactions in transcriptionally active loci from deconvolved single cells. *Genome Biology*, 21(1):1–17, 2020.
- [2] Clemens B Hug, Alexis G Grimaldi, Kai Kruse, and Juan M Vaquerizas. Chromatin architecture emerges during zygotic genome activation independent of transcription. *Cell*, 169(2):216–228, 2017.
- [3] Leslie J Mateo, Sedona E Murphy, Antonina Hafner, Isaac S Cinquini, Carly A Walker, and Alistair N Boettiger. Visualizing dna folding and rna in embryos at single-cell resolution. *Nature*, 568(7750):49, 2019.
- [4] Yuki Ogiyama, Bernd Schuettengruber, Giorgio L Papadopoulos, Jia-Ming Chang, and Giacomo Cavalli. Polycomb-dependent chromatin looping contributes to gene silencing during drosophila development. *Molecular cell*, 71(1):73–88, 2018.